



# Comparing CLE-AdCSV applications using SA and TAC to determine the Fe binding characteristics of model ligands in seawater

Loes J.A. Gerringa[1], Martha Gledhill[2], Indah Ardiningsih[1], Niels Muntjewerf[1], Luis M.Laglera[3]

[1]Royal Netherlands Institute for Sea Research (NIOZ), Department of Ocean Systems, Texel, the Netherlands.
[2]GEOMAR Helmholtz Centre for Ocean Research, 24148 Kiel, Germany.
[3]FI-TRACE, Departamento de Química and ‡Laboratori Interdisciplinari sobre Canvi Climàtic, Universidad de las Islas Baleares, Palma, Balearic Islands 07122, Spain.

*Correspondence to: loes.gerringa@nioz.nl*

**Abstract.** Competitive ligand exchange–adsorptive cathodic stripping voltammetry (CLE-AdCSV) is used to determine the
conditional concentration ([L]) and the conditional binding strength ($\log K^{cond}$) of dissolved organic Fe-binding ligands, which together influence the solubility of Fe in seawater. Electrochemical applications of Fe speciation measurements vary predominantly in the choice of the added competing ligand. Although different applications show the same trends, [L] and $\log K^{cond}$ differ between the applications. In this study, binding of two added ligands in three different common applications to three known types of natural binding ligands are compared. The applications are: 1) Salicylaldoxime (SA) at 25μM (SA25)
and short waiting time, 2) SA at 5μM (SA5) and 3)2-(2-thiazolylazo)-ρ-cresol (TAC) at 10 μM, the latter two with overnight equilibration. The three applications were calibrated under the same conditions, although having different pH values, resulting in the detection window centers (D) DTAC > DSA25 ≥ SA5 (as log D values with respect to Fe3+: 12.3>11.2≥ 11).

For the model ligands, there is no common trend in the results of $\log K^{cond}$. The values have a considerable spread, which indicates that the error in $\log K^{cond}$ is large. The ligand concentrations of the non humic model ligands are overestimated by
SA25 which we attribute to the lack of equilibrium between Fe-SA species in the SA25 application. The application TAC more often underestimated the ligand concentrations and the application SA5 over and under estimated the ligand concentration. The extent of overestimation and underestimation differed per model ligand, and the three applications showed the same trend between the non humic model ligands especially for SA5 and SA25. The estimated ligand concentrations for the humic and fulvic acids differed approximately 2 fold between TAC and SA5 and another factor of 2 between SA5 and
SA25.

The use of SA above 5 μM suffers from the formation of the species Fe(SA)x (x>1) that is not electro-active as already suggested by Abualhaija and Van den Berg (2014). Moreover, we found that the reaction between the electro-active and non-electro-active species is probably irreversible. This undermines the assumption of the CLE principle, causes overestimation of [L] and could result in a false distinction into more than one ligand group.
For future electrochemical work it is recommended to take the above limitations of the applications into account. Overall, the uncertainties arising from the CLE-AdCSV approach mean we need to search for new ways to determine the organic complexation of Fe in seawater.



## 1. Introduction

The trace element Fe is an important micro-nutrient for phytoplankton (De Baar and La Roche, 2003; Achterberg et al., 2018; Lauderdale et al., 2020). Together with light it limits the growth of phytoplankton in 30 to 40% of the oceans (De Baar, 1990; Martin et al., 1990; Rijkenberg et al., 2018; Boyd et al., 2000;). One of the reasons for the limiting role of Fe is its low solubility in seawater, which can be enlarged at least tenfold by complexation with dissolved organic ligands (Liu and Millero, 2002). The organic complexes of dissolved iron (DFe) in the oceans are important since these decrease the inorganic Fe concentration,

and therefore reduce precipitation as Fe(oxy)hydroxides and adsorption onto particles (scavenging). Organic ligands can also be oxidised under the influence of light and reduce Fe(III) into the labile but more bio-available Fe(II) via ligand-metal charge transfer reactions (Barbeau et al., 2001; Barbeau, 2006; Rijkenberg et al. 2006). Furthermore, organic complexation of Fe can be expected to modify Fe bioavailability, although the relationship between DFe speciation and bioavailability appears to be complex (Van den Berg 1995; Hutchins at al., 1999; Shaked et al., 2005; Salmon et al., 2006; Morrisey and Bowler, 2012;

Gledhill and Buck, 2012; Shaked et al., 2020). The significance of Fe speciation to its biogeochemistry has led to incorporation of chemical Fe speciation into global biogeochemical models, with varying levels of complexity (Tagliabue and Völker, 2011, 2015; Ye and Völker, 2017). Recent modelling work has also highlighted the potential importance of the physico-chemical environment on Fe speciation, in particular highlighting the role that pH plays in modifying Fe speciation (Ye et al., 2020). The role of both pH and temperature is potentially of great significance considering climate change and ocean acidification.

Out of the natural ligand pool, the following Fe-binding organic ligands groups have been identified:

1. siderophores, being relatively strong Fe-binding ligands excreted by micro-organisms to bind Fe and make it bio-available (Gledhill et al., 2004; Mawji et al., 2008, 2011; Boiteau and Repeta, 2015; Boiteau et al., 2018);
2. humic substances, a diverse group of large molecules, that include ligands with affinity for iron in a broad range possibly spanning from weak to as strong as some siderophores (Laglera et al., 2007, 2011, 2019, Su et al., 2018; Slagter et al., 2019);
3. polysaccharides, a group of ligands binding Fe relatively weakly (Hassler et al, 2011, 2015), although stronger polysaccharides have also been reported (Norman et al. 2015).

Organic ligands increase the solubility and residence time of Fe. Although specific methods exist that focus on analyzing siderophores, humic materials and polysaccharides, the connection between the actual abundance of these groups and the

overall Fe-binding capacity is often not well resolved. A few exceptions include Laglera et al. (2019) who determined specifically humic Fe-binding ligands, Boiteau et al. (2018) who focused on DFe bound to siderophores and Bundy et al. (2014, 2015) who combined methods to determine the abundance of specific groups.

For approximately three decades, competitive ligand exchange-adsorptive cathodic stripping voltammetry (CLE-AdCSV) has been used to estimate the overall Fe-binding capacity of organic matter in seawater. The technique estimates the conditional

concentration of ligands in the sample ($[L]$), and the conditional stability constant ($K_{FeL}^{cond}$) of their complexes without specifying the different contributions of specific ligands (Gledhill and van den Berg, 1994; Rue Bruland, 1995; Wu and Luther,





1995; Croot Johansson, 2000; Boye et al., 2001; Buck et al., 2007; Cabanes et al., 2020; Ardiningsih et al., 2020). The term "conditional" is extremely important and means that the obtained results are specific to the composition of the sample matrix analysed (DFe, temperature, pH, ionic strength). Since $[L]$ also depends on the conditions like pH, salinity and dissolved

organic matter, we will use the term conditional for both parameters. The results cannot therefore be considered as an absolute quantification of the properties of all the available Fe-binding sites and extrapolated to other conditions or matrices (Gledhill and Gerringa, 2017, Town and van Leeuwen, 2014). The technique uses an added ligand (AL) with known concentration and conditional stability constant with Fe that form an electro active Fe-ligand complex that competes with the natural ligands present in a sample for Fe. The sample is equilibrated for a defined time period under controlled conditions of pH, light and

temperature. The Fe bound to the artificial ligand is analyzed through its electro-active properties. By adding increasing amounts of Fe to subsamples, the competing natural organic ligands are titrated until the natural binding sites are no longer strong or abundant enough to compete successfully with the AL. The competition is reflected by the increased proportion of Fe bound to the artificial ligand and from this the conditional concentration and binding strength can be calculated (Van den Berg 1982). Although the method does not provide information on the molecular composition of the binding sites, CLE-

AdCSV does give information on the Fe binding capacity of seawater at the measurement pH, temperature and DFe concentration of the sample. Thus, an indication of the potential capacity for further Fe-binding in a particular sample can be assessed (van den Berg 1995; Tagliabue and Völker, 2011; Pham and Ito, 2019). Application of CLE-AdCSV allowed in some samples the division of the overall ligand in two broad ligand groups as a function of their conditional stability constants indicated with 1, ($K_{1,FeL}^{cond}$), for the relatively strong ligand group and with 2, ($K_{2,FeL}^{cond}$), for the relatively weak ligand group (Rue

and Bruland, 1995, 1997, Buck et al 2015, 2018; Bundy et al., 2014, 2015).

Four different ALs have been reported as forming effective electroactive complexes for the purposes of CLE-AdSCV: 1-nitroso-2-napthol (NN) (Gledhill and van den Berg, 1994), salicylaldoxime (SA) (Rue and Bruland 1995), TAC 2-(2-thiazolylazo)-p-cresol (Croot and Johansson, 2000) and 2,3-dihydroxynaphthalen (DHN) (Van den Berg, 2006). The two ALs, SA and TAC, are the usual selection in field studies (Rue and Bruland, 1995, 1997; Croot and Johansson, 2000; Croot et al.,

2004; Boye et al., 2005; Thuróczy et al., 2011a, 2011b, 2012; Kondo et al., 2012; Bundy et al., 2014, 2015; Buck et al., 2015, 2018; Gerringa et al., 2015, 2017; Abualhaija et al., 2015; Klein et al., 2016; Slagter et al., 2017, 2019) and basin scale data sets now exist for $K_{FeL}^{cond}$ and $[L]$ obtained using these two ALs (Caprara et al., 2016; Cabanes et al. 2020; Schlitzer et al., 2018) which provide an important resource for our understanding of iron biogeochemistry in the ocean (Boyd and Ellwood, 2010; Boyd and Tagliabue, 2015; Völker and Tagliabue, 2015; Tagliabue et al., 2016; Lauderdale et al., 2020). However, results of

inter-comparisons of field data suggest that although trends may be similar for different ALs, the different methods may not be directly intercomparable as conditional $[L]$ differed significantly, with SA giving higher $[L]$ and often identifying more than one ligand group compared with TAC (Buck et al., 2012; Buck et al., 2015). With SA, often two ligand groups can be distinguished, while TAC distinguishes only one, complicating comparison of trends in $K_{FeL}^{cond}$. The $K_{FeL}^{cond}$ obtained by TAC is in between the two $K_{FeL}^{cond}$ values of the two groups obtained with SA. The question is therefore - what is the underlying cause



of these differences? It was found urgent within the SCOR work group 139 to test or calibrate methods with model ligands. Although there are studies that determined $[L]$ and $K_{FeL}^{cond}$ for model ligands such as siderophores, with some success with respect to $[L]$ at least (Rue and Bruland, 1995; Buck et al. 2000; Witter e al. 2000; Croot and Johansson, 2000), a thorough examination of multiple ligands and approaches that also sought to compare determined $[L]$ and $K_{FeL}^{cond}$ with values calculated from thermodynamic constants has not been previously undertaken to our knowledge.In this work we chose to examine

potential bias between ALs via a series of carefully controlled studies of selected Fe-binding ligands that are likely representative of those found in the marine environment. We chose to work with SA and TAC, and further compared two reported SA methods. Our three approaches comprised a) 10 µM TAC with overnight equilibration at pH = 8.05 (Croot and Johansson, 2000), b) 25 µM SA (SA25) at pH = 8.2 with a short waiting time of 20 minutes for the competing reaction to occur (Rue and Bruland 1995, Buck et al., 2007), and c) 5 µM SA (SA5) at pH = 8.2 with overnight equilibration as described

by Abualhaija et al., (2015). Since all parameters derived in CLE-AdCSV are fundamentally dependent on the side reaction coefficient of the Fe-AL under the conditions of analysis, we calibrated each ligand in the same laboratory under comparable conditions for consistency and to avoid any issues of bias relating to the choice of calibrating ligand, the calculation methods employed in the original papers and the choice of side reaction coefficient for Fe. We made some (arbitrary) choices on conditional binding constants between DTPA and Fe, however, we worked with one set of thermodynamic constants to make

comparison between the methods consistent. We therefore press the point that the focus of the paper is on comparing the empirical outcome of the three applications and not on the accuracy of $K_{FeL}^{cond}$.

We used Diethylenetriaminepentaacetic acid (DTPA) as a model simple well-defined molecule, the naturally occurring phytic acid, the hydroxamate siderophores desferrioxamine B, ferrioxamine E, ferrichrome, and the catecholate siderophore virbriobactin, and fulvic and humic acids. Moreover, we carried out for the first time a specific study on the kinetics of complex

formation and ligand exchange of Fe(SA)x complexes. All the experiments were performed in a single laboratory in order to minimize inter-laboratory variations in protocol, material and reagent variations. We begin with a short review of previous criticisms of the CLE-AdCSV approach, since this provides important context for our study. Our overall aim was to shed light on the processes that lead to method discrepancies in the determination of natural iron ligand concentrations.

## 2. Potential origins of bias in the determination of binding parameters by CLE-AdCSV

CLE-AdCSV is based on many limitations and assumptions which have been discussed at some length in the literature (e.g. Apte et al., 1988; Turoczy and Sherwood, 1997; Town and Filella, 2000; Hudson et al., 2003; Croot and Heller, 2012; Laglera et al., 2013; Town and van Leeuwen, 2014; Gerringa et al., 2014; Laglera and Fillela, 2015; Pižeta et al., 2015; Turner et al., 2016; Gledhill and Gerringa, 2017). If the assumptions are sufficiently satisfied, the calculation of ligand complexation parameters like the conditional ligand concentration $[L]$and the conditional stability constant $\log K_{FeL}^{cond}$ can be undertaken,

usually using the Langmuir isotherm (e.g. Gerringa et al., 2014 and references herein).

Here we give a brief overview of the limitations and assumptions.





Thermodynamic equilibrium between Fe, the added ligand and natural ligands must be established. Failure to achieve equilibrium can lead to incorrect estimates of $[L]$ and the conditional constants (Hudson, 1998; Gerringa et al., 2014; Town and van Leeuwen, 2014; Laglera and Filella, 2015). Non-equilibrium conditions arise if the electroactive complex is a reaction

intermediate, if insufficient time is allowed for equilibration of the reactants, or if there is a fraction of DFe that is kinetically inert in the time scale of the equilibrium period (e.g. aged inorganic colloids or Fe(AL) complexes).

There must be a detectable level of competition between the added and natural ligands. The competitive interaction is summarised by the side reaction coefficients for the natural and added ligands ($\alpha_{FeL}$, $\alpha_{FeAL}$ respectively). The side reaction coefficient, which is often expressed as a logarithm, is defined as

$$\alpha_{FeL} = K_{FeL}^{cond} \times [L'] = \frac{FeL}{Fe'} \qquad\qquad (1)$$

or

$$\alpha_{FeAL} = K_{FeAL}^{cond} \times [AL'] = D \qquad\qquad (2)$$

where $[L']$ and $[AL']$ are the conditional concentration of the organic ligand not bound by Fe and the concentration of AL not

bound by Fe respectively and $Fe'$ is the Fe concentration not bound to L. The side reaction coefficient of AL defines the center of the detection window or analytical window, which we defined here as D to prevent confusion between side reaction coefficients of added and natural ligands. The window is assumed to be approximately two to three orders of magnitude wide, one to two orders of magnitude above and below D (Apte et al. 1988; Van den Berg and Donat, 1992; Milller and Bruland, 1997, Laglera et al., 2013; Laglera and Fillela, 2015). In practice, the upper limit of D is defined by the analytical sensitivity

of the AdCSV method, as it is bound by the limit of detection of FeAL. The lower limit of D is bound by ligands that are outcompeted by the AL within the range of Fe added during the titration. Since AdCSV is internally calibrated via standard additions, in practice the lower limit of the detection window is bound by the value of $\alpha_{FeL}$ achieved when the analytical response is deemed to be linear (Apte et al, 1988; Laglera and Fillela, 2015).

The concentration of the FeAL complex can be accurately determined at each titration point. AdCSV is internally calibrated

via standard additions, D and values obtained for $K_{FeL}^{cond}$ and $[L]$ are strongly influenced by our ability to accurately calculate the sensitivity (Turoczy and Sherwood, 1997, Hudson et al., 2003; Pizeta et al., 2015).

Complexes of Fe with natural ligands cannot be electro-labile under the experimental conditions since this could result in interferences with the actual detection of the Fe-AL complex (Yang and van den Berg, 2009; Laglera et al., 2011).

The equilibrated Fe-AL complex must be electro-active since it is the reaction on which the detection is based.

The AL should not react with the natural ligands altering or cancelling their binding ability.

In the last decade, many studies have questioned the compliance of the CLE-AdCSV methodology to these assumptions and their influence on method discrepancies. Laglera et al. (2011) showed the inability of TAC to measure fulvic and humic acids as Fe-binding dissolved organic ligands, which might be due to either assumption 2 or 6. Humic substances are ubiquitous,



they form large diverse molecules and are broadly recognized as Fe-binding ligands (Krachler et al, 2015; Su et al., 2018; Laglera et al 2019; Whitby et al., 2020, Yamashita et al., 2020). According to other work, TAC is able to detect at least part humics as Fe binding organic ligands (Batchelli et al., 2010; Slagter et al., 2017; Dulaquais et al., 2018).

The SA25 application has been criticized for not meeting assumption 1, SA25 has a waiting time of 15 to 20 minutes in contrast with the overnight equilibration used for the TAC and SA5 applications (Abualhaija and Van den Berg, 2014, Abualhaija et

al., 2015; Slagter et al., 2019).

Abualhaija and Van den Berg (2014) found that two Fe-SA complexes are formed, FeSA and Fe(SA)$_2$, and only FeSA is electro-active. At higher [SA], the proportion of Fe(SA)$_2$ increases and the analytical signal decreases resulting in a negative relationship between sensitivity and competitive force. Finally, we would like to point out that the pH of the analysis may have a larger influence on the organic complexation of DFe than previously thought (Gledhill et al, 2015; Avendaño et al., 2016;

Ye et al., 2020). And the same competition of OH ions in binding Fe, irrespective of the buffered pH values of the SA and TAC applications is sometimes used (8.2 and 8.05, respectively, which is a factor 1.4 different in terms of H$^+$ concentration). This complicates a direct comparison of data even more.

## 3. Methods

The natural seawater used in the experiments consisted of mixed leftover samples of the northern Western Atlantic GEOTRACES cruise GA02 (stored frozen). A sample volume, assumed to be necessary for the following few days, was thawed, mixed, UV-irradiated to destroy the natural organic Fe-binding ligands, and stored in the refrigerator. Consequently, one batch differs from others in DFe content. Samples for DFe analysis were taken from every batch. UV irradiated sea water was stored for 3 days at most.

### 3.1 Equipment and measuring conditions

### 3.1.1. Equipment and electrochemical parameters

We carefully followed procedures as described in the literature to ensure methodology was as close as possible to that originally described (Tables S1 and S2). Three different voltammetric setups were used (Table S1). A standard Metrohm set up was used for TAC. For use with SA, a separate Metrohm system was modified to allow for air purging whilst the mercury drop formation

was still executed under nitrogen pressure. Nitrogen did not leak into the headspace of the sample during the measurements in our Metrohm stand. However, when drops are formed pulses of nitrogen are released and end up in the headspace of the sample, purging with air would remove (at least part of) the nitrogen. To check a potential effect of this nitrogen a kinetic experiment with SA25 was executed. To five identical subsamples SA was added at the same time. These subsamples were each measured repetitively during one to several hours, one after the other. No effect was seen after sub sample replacements

(Figure S1). We concluded that nitrogen from the stand did not influence the kinetic process, since the measured FeAL





concentrations had a gradual change over time, independent of the subsample. For the other kinetic experiments with SA, BASi equipment was used (Table S1). The electrochemical settings used by Croot and Johansson (2000), Buck et al. (2007) and Abualhaija and van den Berg (2014) were used without alteration and are summarized in Table S2.

### 3.1.2 Conditioning and equilibration

Electroactive complexes with Fe typically have low solubility and thus tend to adsorb on the walls of containers. Conditioning and equilibration of all contact surfaces is thus an important pre-treatment step in order to minimize losses of Fe and ligand species during the course of the experiment. Different materials do not have the same adsorption properties (Fischer et al., 2006), all cells and titration vials were made of Teflon, other bottles, sample bottles and those used for kinetic experiments were low density polyethylene (LDPE) bottles (Nalgene[TM], Fisher Scientific). For all three applications, the same materials were used, cancelling any deviation among methods from the interaction of solution component and containers.

Before use, all materials such as vials, bottles and cells, were conditioned overnight with the prepared combinations of seawater and ligand. The conditioning procedure was performed at least three times for the analysis with TAC and at least five times for analysis with SA. The cell with electrodes, stirrer and purge tube were kept overnight in low metal seawater. Before a titration started first two measurements were executed with seawater containing all chemicals but no Fe addition, these measurements served also as check for possible contamination of the cell. Hereafter, two zero additions were measured (see section 3.4), of which the second was used as start of the titration.

Before starting kinetic measurements, a 30 ml vial with the same content and treatment as the sample was used for three analyses (thus three times 10 ml) in order to condition the cell wall, electrodes and stirrer.

The 200 ml bottles used for kinetic studies were conditioned with 6 nM Fe, in the absence of the added ligand. For tests with UV irradiated seawater without a model ligand, 200 ml bottles were conditioned by rinsing the bottle three times for two minutes with 30 ml of the test seawater. Since UV-irradiated seawater did not contain Fe-binding organic ligands, most of the added 6nM DFe would adsorb on the bottle walls or precipitate.

Samples were equilibrated according to the specific method descriptions, which was overnight equilibration for the TAC and 5 µM SA and 15 minutes for 25 µM SA (Croot and Johansson, 2000, Buck et al., 2007; Abualhaija and Van den Berg, 2014). The 15 minutes equilibrations were applied precisely using a stopwatch, whereas overnight equilibration resulted in a period of at least 14 hours.

### 3.1.3 UV irradiation

Samples, without any additions, were poured into 30 ml Nalgene FEP bottles and placed in in a custom-made UV box between 4 TUV 15W/G15 T8 fluorescent tubes (Phillips) for 4 h. These bottles were tested in the past for photo-oxidation of organics by Co analyses by ICPMS and Cu ligand titrations and proved to be UV permeable enough to destroy organic material.



Precipitates were not observed. After UV irradiation, samples were transferred into a trace-metal clean one-liter LDPE bottle and kept in the refrigerator.


### 3.1.4 Model Ligands

The following discrete synthetic ligands of known concentration (model A ligands) were used at a concentration of 2 nM, unless otherwise stated. Humics (model B ligands, 0.1 or 0.2 mg/liter) were added in a concentration to give an iron binding capacity of approximately 3 nM (Laglera and Van den Berg, 2009; Yang et al., 2017; Sukekava et al., 2018). The stoichiometry

of the formed Fe-model ligand complexes differs for each model ligand. In order to simplify the comparison of binding strengths, stability constants are given for a 1:1 stoichiometry:

Model A ligands

- Diethylenetriaminepentaacetic acid (DTPA $C_{14}H_{23}N_3O_{10}$, Sigma-Aldrich D6518-5G). DTPA was used to calibrate the added ligands via reverse titration according to methods described previously (Croot and Johansson,

2000). We calculated a conditional binding constant $Log K^{cond}_{FeDTPA,Fe3+}$ of 19.01 using the ion pairing speciation software visual MINTEQ (Gustafsson, 2012), disregarding the formation of FeOHDTPA. The $Log K^{cond}_{FeDTPA,Fe3+}$ value was independent of the pH difference 8.05-8.2. This is 0.34 higher than the value (18.65) used by Croot and Johansson (2000) at I=0.7, pH =8.05 with the difference most likely arising as a result of the lower ionic strength predicted by the ion-pairing model.

- Phytic acid ($C_6H_{18}O_{24}P_6 \cdot xNa^+ \cdot yH_2O$, Sigma 68388). According to Witter et al., (2000), $Log K^{cond}_{FePA,Fe3+}$= 22.3-22.4. Rijkenberg et al. (2006) warned that at high phytic acid concentrations aggregates are formed, but at our concentrations (2 nM) this should not be a problem.

- The hydroxamate siderophore desferrioxamine B ( $C_{25}H_{48}N_6O_8 \cdot CH_4SO_3$, has a thermodynamic stability constant of 30.5 (I=0.1; Hider and Kong, 2010). According to Witter et al. (2000) the conditional stability constant,

$log K^{cond}_{FeDFOB,Fe3+}$, is between 21.6 and 22.1 (I=0.7). Van den Berg (2006) found $log K^{cond}_{FeDFOB,Fe3+}$=21.5, whereas Croot and Johansson(2000) concluded that this conditional stability constant was too high and outside D of their TAC method ( $log K^{cond}_{FeDFOB,Fe3+}$>23.4). However, new side reaction coefficients of major cations have been determined since, which give rise to a $log K^{cond}_{FeDFOB,Fe3+}$ of 24.3 at seawater salinity (Schijf and Burns, 2016). (Novartis RVG03984 U.R., 477881 NL.).

- The hydroxamate siderophore ferrichrome $C_{27}H_{45}N_9O_{12}$. Hider and Kong (2010) gave a $log K_{FeL,Fe+}$= 29.1 (I=0.1). According to Witter et al. (2000), $log K^{cond}_{FeL,Fe3+}$ in seawater varies between 21.6 and 22.9 depending on the applied method. Kinetic measurements determining formation constant resulted in 22.9, the equilibrium approach with Fe-titration resulted in 21.6. (Ferrichrome Iron-free from *Ustilago sphaerogena* Sigma Aldrich (F8014-1MG)).



- The hydroxamate siderophore ferrioxamine E, $C_{27}H_{45}FeN_6O_9$. According to Hider and Kong (2010) Ferrioxamine E has a higher affinity for Fe than ferrioxamine B ($\log K_{FeL,Fe3+}$= 32.5, at I=0.1). However, no information of the conditional stability constant in seawater is known for ferrioxamine E. This model ligand as purchased was already saturated with Fe. (Ferrioxamine E from *Streptomyces antibioticus* Sigma Aldrich (38266-3MG-F).

- The triscatecholate siderophore Vibriobactin ($C_{35}H_{33}FeN_5O_{11}$). No information is available on the Fe binding characteristics of this model ligand, but in general catecholates have higher binding strengths with Fe than hydroxamates because of their ortho phenolate binding groups (Hider and Kong, 2010). (Vibriobactin (iron-free) from *Vibrio cholerae* V69, EMC micro collections).

Model B ligands

Humic substances are the heterogeneous mix of hydrophobic compounds originating from chemical and microbial transformation of living matter as it decays in the environment.

- Fulvic Acids (FA) is the smaller and more soluble fraction of humic substances (Buffle, 1990). Therefore, this is not just the ligand but a series of compounds of which a fraction functions as iron binding ligands. Laglera and Van den Berg (2009) determined that 1 mg of this specific FA binds 16.7 ± 2.0 nM Fe with $K_{FeL,Fe'}^{cond}$=10.6. (IHSS Suwannee River Fulvic Acid Standard II, 1R101F). Whereas Yang et al (2017) and Sukekava et al., (2018) found that 1 mg could bind 14.6±0.7 nM Fe. (IHSS Suwannee River Fulvic Acid Standard II 2S101F). It must be noted that the batches are different between the above results and these values can differ per produced batch, still we assumed 2.92 nM Equivalents (nM Eq) of ligand sites to be added with 0.2 mg SRFA per liter.

- Humic acids (HA) are the larger fraction of humic substances that precipitate at low pH (pH 2) (Buffle, 1990). Laglera and Van den Berg (2009) determined that 1 mg of this specific HA binds 32 ± 2.2 nmol Fe with $\log K_{Fe'L}^{cond}$=11.1. (IHSS Suwannee River Humic Acid Standard II, 2S101H). We assumed that 0.1 mg added HA per liter would add 3.2 and 0.2 mg 6.4 nM Eq of ligand sites.

**3.2 AL Calibration**

Seven or eight conditioned Teflon 30 ml vials were filled with 10 ml UV irradiated seawater spiked with buffer, 6 nM Fe (Table S1) and increasing amounts of the calibrating ligand DTPA. For TAC the pH was 8.05 and for SA the pH was 8.2 according to the original method specifications. The calibrations were repeated 4 times. The buffer used for all applications was ammonium borate (Abualhaija and van den Berg, 2014; Buck et al., 2007). Details can be found in the Supplementary Information.





DTPA additions were 0, 10, 100, 200, 400, 1000, 2500 nM DTPA for TAC, 0, 1, 10, 40, 80, 100 and 200 nM DTPA for SA5 and 0, 1, 40, 100, 200, 400 and 1000 nM DTPA for SA25. Mixtures of UV seawater with buffer, DFe and DTPA were equilibrated for at least 8 h after which SA or TAC were added. New mixtures were equilibrated either overnight or for 15 min in the case of SA25, after which peak heights for FeAL were determined following the procedures described in Table S1. We

calibrated SA25 using a short waiting time instead of 5 h of equilibration (Rue and Bruland, 1995; Buck et al., 2007) to ensure consistency with the approach applied to samples. For the calibration of TAC the normal Metrohm instrument was used, for SA the BASi instrument (Tables S1, S2). The measurements were done in sequence of increasing DTPA concentrations, without rinsing cells in between.

### 3.3 Signal stability tests

As we found the CSV signal after SA addition lacked stability and decreased with time, we performed a series of experiments to find the cause. We tested the following aspects for both instruments: the influence of a purge step with air (Figure S2), the influence of the size of the mercury drop (Figure S3, Table S3), the influence of the mercury puddle on the bottom of the cell, and the influence of the SA concentration. For the kinetic measurements (section 2.4) of the SA applications, both BASi and Metrohm stands were used. Details on procedures are given in the Supplementary Information file.

### 3.4 Titrations

### 3.4.1 TAC

Fifteen vials were prepared with increasing Fe content, in a mixture of UV-irradiated seawater and model ligand (Table S1, Croot and Johannsson, 2000, Ardiningsih et al., 2020). Blanks were obtained by analysis in the absence of model ligands.

### 3.4.2 SA5

The application followed Abualhaija and van den Berg (2014), but used the above described BASi instrument. SA (added to a final concentration of 5 µM), buffer, Fe additions (Table S1) and samples were left to equilibrate overnight.

### 3.4.3 SA25

For SA25, 25 µM SA buffer and DFe were added one hour before analysis. SA was added to a final concentration of 25 µM separately to each vial, 15 minutes before the measurement.


### 3.5 Kinetic measurements

The samples contained either UV-irradiated seawater, or UV-irradiated seawater with a model ligand to which buffer and 6 nM Fe were added in a pre-conditioned bottle. If a model ligand was present, this was first allowed to equilibrate overnight





with the buffer and 6 nM Fe. In samples with only UV-irradiated seawater, two approaches were followed: One in which Fe

was added together with TAC or SA at t=0, and one in which Fe was equilibrated overnight prior to addition of TAC or SA. In the latter case, there is the possibility that Fe-oxide precipitates were formed and dissolved after addition of TAC or SA. At t = 0, TAC or SA was added.

At t = 0, the first measurements were done as rapidly as possible until approximately t = 1 h, followed by subsequent measurements every 20 min, every 40 min and 1 h until either t = 4 h or t = 7 h. The number of analyses depended on the

application and experiment duration (4 or 7 h) but contained a minimum of 14 duplicate measurements. In this way the FeAL$_{(x)}$ formation in time can be followed. However, the model ligand dissociation rates cannot be calculated from the rate of peak increment because the addition of Fe (6 nM) was in excess of the model ligand concentration (2 nM Eq, if not indicated differently). Therefore, the excess Fe formed hydroxides and adsorbed on the cell and electrode surfaces. The increment of signal reflects the competition of TAC with all these iron species and not just with the model ligand.

Two protocols were followed:

1. In-cell experiments: repeated scans of the same sample. In this way, contamination was prevented and more measurements could be undertaken, especially at the start of the experiments. The total time of the experiments lasted 4 or 7 h. Samples of 30 ml were prepared of which the first 20 ml were used to condition the cell twice, after which, the last 10 ml was transferred to the cell and the experiment undertaken. The AL was added to the cell at t

= 0. For TAC, the addition took place after the purge step to reduce the time-lapse between addition and first measurement. An extra set of in-cell experiments were carried out with UV irradiated seawater, natural seawater and UV irradiated seawater spiked with DTPA, 40 nM for SA=AL and 200 nM for TAC=AL. Other experiments for all model ligands were repeated with 2 nM of added model ligand. In this protocol, mercury accumulated in the cell during the experiment.

2. Bottle experiments: scans were carried out on separate aliquots of one sample. In this experiment, a fresh aliquot of 10 ml was pipetted into the voltammetric cell for the determination of peak height at each time point. The total sample volume was 200 ml and the AL was added at t = 0. In this experiment, accumulation of mercury at the bottom of the cell was limited. The experiment lasted for 4 or 7 h, consistent with the "in-cell" approach. The first 10 ml was transferred as quickly as possible into the preconditioned cell and the measurement started. In order to

determine the amount of adsorbed Fe on the 250 ml bottle walls, the bottles were rinsed carefully with 5 ml of elution acid (1.5 M teflon distilled HNO$_3$ that contained rhodium see below section 2.6 ICPMS analysis) and Fe concentration in the acid rinse determined by ICPMS (section 3.7).





### 3.6 Calculations

The sensitivity, S, the ligand concentration [L] and conditional stability constant ($K^{cond}$) were calculated by direct non-linear fitting of the Langmuir isotherm (Gerringa et al., 2014) with inherent co-dependence of [L] and $K^{cond}$ (Apte et al., 1988; Hudson et al., 2003; Gerringa et al., 2014)

The inorganic side reactions of DFe with dissolved hydroxides, $\alpha_{Fe'}$, was calculated using the constants from Liu and Millero (2002) resulting in an inorganic alpha for Fe ($\alpha_{inorg}$) of log$\alpha_{inorg}$ = 9.9 at pH = 8.05 and of log$\alpha_{inorg}$ = 10.4 at pH = 8.2. These are

slightly different from literature values of Croot and Johansson (2000) and Abualhaija and van den Berg (2014). The conditional binding strength of DTPA was obtained using Visual Minteq. We used an average seawater major ion composition, and an average deep sea DFe concentration of 0.5 nM was chosen for these calculations. DTPA was added to the composition at the concentrations used, and the pH was fixed at values of 8.05 and 8.2. According to the VMinteq calculations, $K_{FeDTPA,Fe3+}$ = 27.3, the logarithm of side reaction coefficients for DTPA with major cations was 8.26, resulting in

$K^{cond}_{FeDTPA,Fe3+}$ = 19.01.

### 3.7 ICPMS analysis of dissolved Fe

Dissolved Fe was analysed with a Thermo Finnigan HR-ICPMS element 2 (for details see Middag et al., 2015; Gerringa et al., 2020). Briefly, seawater aliquots, with and without the addition of model ligands were concentrated using a seaFAST system

after UV-destruction. Background Fe concentrations in TAC, SA and both buffers were determined by pipetting 100 µl in 20 ml of elution acid (1.5 M Teflon distilled HNO$_3$ containing rhodium). These samples were measured by ICPMS without further sample handling as were the acid rinse samples to measure adsorption on bottle walls. The results of the ICPMS on samples with added ligands is given in Table 2 as the DFe of the samples. Upon addition of the buffers 0.04 nM DFe was added inadvertently to the samples. The addition of 10 µM TAC added 0.2 nM Fe, 25 µM SA 0.2 nM and 5 µM SA 0.04 nM Fe.

These inadvertent additions have been included in the Fe concentrations. The acid rinse of the 250 ml LDPE bottles, contained 106 nM Fe, when conditioned with 6 nM Fe and TAC and 59 nM Fe when conditioned by 6 nM Fe, 2nM phytic acid and TAC. This means that the potential release in a 200 ml sample could be at maximum 1.3 and 0.7 nM Fe. The difference in Fe adsorption on the bottle wall shows very well the effect of conditioning.

### 4 Results and discussion






### 4.1 Calibration of Fe-AL α coefficients

Details of the calibration are given in the Supplementary Information, here we explain our choice to use an overall α coefficient for SA as AL instead of the sum of separate α coefficients of FeSA and FeSA2. We further present and discuss the resulting binding characteristics of the AL's.

The competition by DTPA causes a reduction in peak height compared to the situation without DTPA (Figure S4). At equilibrium, dissolved Fe is distributed over the following species,

$$DFe = [FeDTPA] + [FeAL_2] + [FeAL] + [Fe'].\qquad(3)$$

For the application of TAC, the contribution of FeTAC is thought to be negligible with respect to the formation of Fe(TAC)$_2$ (Croot and Johansson, 2000). For SA in the micromolar range, both FeSA$_2$ and FeSA are formed, although only FeSA is the

electroactive species (Abualhaija and Van den Berg, 2014). Using (1) this gives,

$$DFe = \alpha_{FeDTPA} \times [Fe^{3+}] + \alpha_{FeAL2} \times [Fe^{3+}] + \alpha_{FeAL} \times [Fe^{3+}] + \alpha_{inorg}[Fe^{3+}] =$$

$$[Fe^{3+}] \times (\alpha_{FeDTPA} + \alpha_{FeAL2} + \alpha_{FeAL} + \alpha_{inorg}).\qquad(4)$$

The α coefficients determine the distribution of Fe over the complexes with DTPA and AL. When [FeDTPA] = Σ[FeAL], the α coefficients of DTPA and AL are equal illustrating that a calibration is actually comparing α-values of the added ligand (AL)

and the calibrating ligand. From the α-values at a determined AL concentration, $K_{FeAL}^{cond}$ and/or $\beta_{FeAL2}^{cond}$ are calculated. The calculation of $\beta_{Fe(SA)2}^{cond}$ cannot be done with precision using only our two SA concentrations. Since we actually need the α-values for calculating the ligand characteristics from the titration data, we do not need to calculate $\beta_{Fe(SA)2}^{cond}$. The α-values include the contributions of $K_{FeAL}^{cond}$ and $\beta_{FeAL2}^{cond}$ (Table 1) (for more detail see SI).

The obtained α values (Table 1) differ from the original literature values, which is likely due to the toolbox we used, Visual

Minteq. If we consider the $\log\alpha_{FeAL,Fe3+}$ values, the calibration results for TAC and SA5 we obtained (Table 1) compare very well with values from the literature (Croot and Johansson, 2000; Abualhaija and Van den Berg, 2014). Our $\log\alpha_{\Sigma FeSA,Fe3+}$ of SA25 (considering both FeSA and Fe(SA)$_2$ formation) shows a larger discrepancy with Buck et al., (2007) than the above comparisons (our $\log\alpha_{\Sigma FeSA,Fe3+}$ = 11.2 versus $\log\alpha_{FeSA,Fe3+}$ = 11.9 of Buck et al. 2007).

The difference becomes larger when calculated with respect to inorganic Fe (Fe′) when using the pH adjusted values of $\log\alpha_{inorg}$

= 9.9 for 8.05 and $\log\alpha_{inorg}$ = 10.4 for pH = 8.2 (Liu and Millero, 2002). For SA5 and SA25, the comparison between our data and literature values is thus offset with respect to Fe′, due to the application of $\log\alpha_{inorg}$ = 10.4. It is possible that the larger deviation in SA25 from previously reported values is partly due to the shorter waiting time used in our study,15 minutes instead of 5 hours (Rue and Bruland, 1995). However, the calibration should be executed according to the published protocol of the analyses.




## 4.2 Titrations

We present the concentrations of FeAL determined during the titrations of the selected ligands with the 3 different methods to allow direct comparison between the approaches (Figure 1, Tables 2 and 3).

Overall $K^{cond}$ and $\alpha_{FeL}$ values of the model ligands were highest with TAC. In other words, they were highest with the

application with the highest D. For model A, ligands like siderophores this points to bias in the determination of $K^{cond}$ perhaps as a result of true values too high to be measured with accuracy. For example, an estimate of 24.25 for $K^{cond}$ for FeDFO in seawater can be calculated using the side reaction coefficient of 6.25 for DFO binding to Ca and Mg at pH 8.0 (Schijf and Burns, 2016; Wuttig et al., 2013) and the stability constant for FeDFO given by Hider and Kong (2010). For the complex ligands, model B, like humic and fulvic acids, which contain multiple binding sites with a range of affinities, an average $K^{cond}$

will be determined based on D of the method being applied (Tables 1 and 2). Both factors highlight important concepts that relate to the CLE-AdCSV approach in general, that need to be taken into consideration when interpreting $K^{cond}$ derived from CLE-AdCSV titrations.

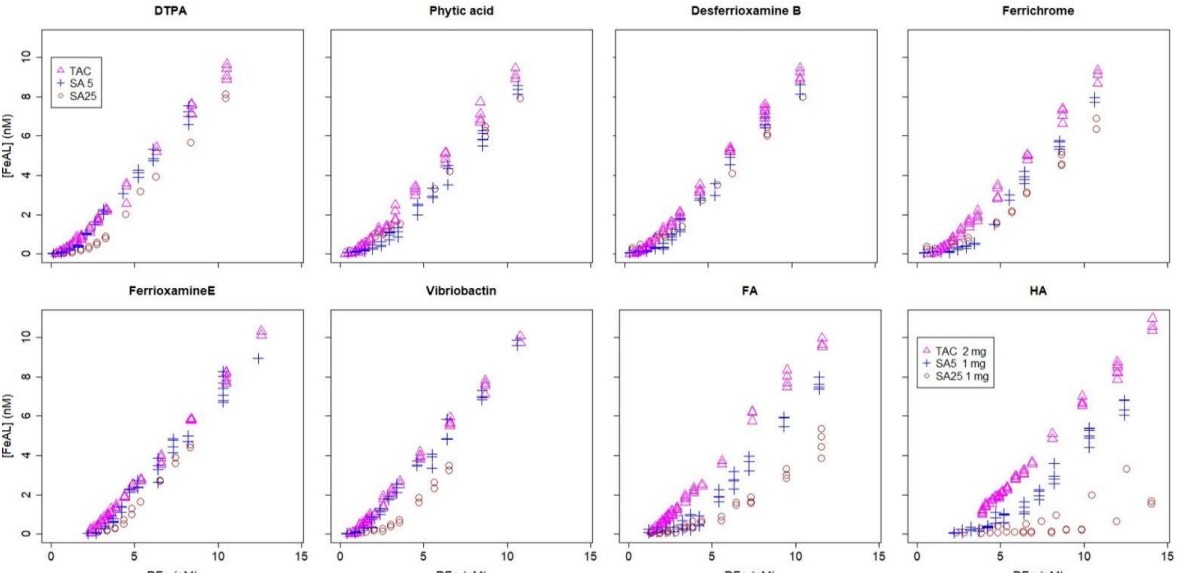

**Figure 1: Iron titrations of UV irradiated seawater containing model ligands in competition with TAC, SA5, and SA25 as added**
**ligand. [FeAL] = Fe-added ligand complex using sensitivity (S)=1, [Fe] = total iron concentration. See Table 2 for the DFe at zero addition. A=DTPA, B=phytic acid, C=desferrioxamine B, D=ferrichrome, E=ferrioxamine E (saturated with Fe), F=vibriobactin, G=fulvic acid FA, H=humic acid HA. Data for TAC and SA5 are from duplicate experiments, for SA25 from single experiments, except for FA and HA where for SA25 also duplicate experiments were done. Note the different HA concentrations, 0.1 and 0.2 mg..**

Ligand concentrations were highest with SA25 and lowest with TAC (Tables 2 and 3, Supplementary Figure S5) and thus

showed the opposite trend to $K^{cond}$. Comparison with the actual added concentrations of the model A ligands shows that [L]



was, with the only exception of the Fe saturated Ferrioxamine E, relatively underestimated by TAC (5-58%) and systematically overestimated by SA25 (26-125%, Supplementary Figure S5). The overestimation by SA25 might be due to a lack of equilibrium. In theory when Fe binding ligands are not yet in equilibrium with the AL, the dissociation of FeL complexes required to reach equilibrium is incomplete and the so-called straight part is curved and not straight. In principle this will

underestimate the ligand concentration. The overestimates observed for SA25 might therefore be caused by disequilibrium in the Fe-SA species. The extent of overestimation and underestimation differed per model ligand (see below), and the three applications showed the same trend between the model A ligands (Supplementary Figure S5) especially for SA5 and SA25. Assuming the concentration of ligand sites per weight unit determined by Laglera and van den Berg (2009) and Sukekava et al. (2018) to be correct, the overestimation by SA25 was larger for model B ligands (Tables 2 and 3). The standard deviation

for duplicate measurements of [L] was 0.3 and 0.2 nM Eq of Fe for the TAC and SA5 application, respectively (excluding Ferrioxamine E because it was saturated with Fe, see below). The standard deviation with SA25 (N=5) was 0.3 nM Eq of Fe. In the following we will assume ±0.3 nM Eq of Fe as precision for [L]. The differences between the applications are smaller when the $\alpha_{FeL}$ values are compared (Table 3), which is understandable, since it is $\alpha_{FeL}$ that is titrated and also because $\alpha_{FeL}$ is calculated from the product of L not bound by Fe ([L′]) and $K^{cond}$ and thus compensates for any codependence between $K^{cond}$

and [L].

We 'back' calculated the titration curves using our present results, $K^{cond}$ and [L], and presented this in log-log plots of [FeAL] versus total dissolved Fe together with the actual data points (Figure 2). For DTPA and desferrioxamine B we added the theoretical titration curves that should be obtained given the $K^{cond}$ calculated from the thermodynamic constants and the 2 nM of added model A ligands. We presented the back calculations in log-log plots in order to magnify the initial part of the

titration (Figure 2).



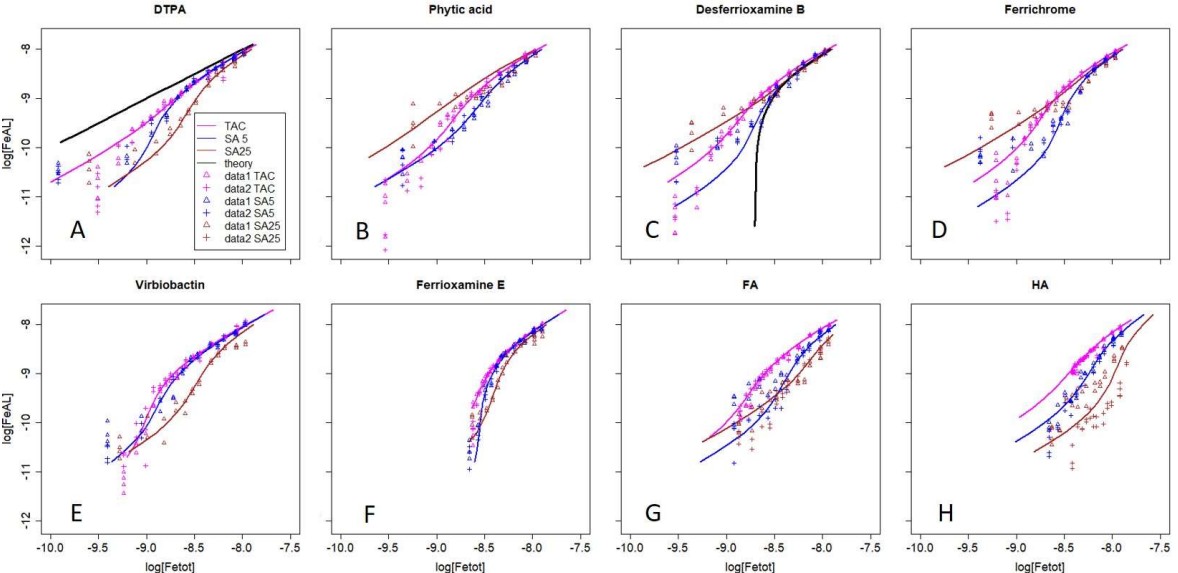

**Figure 2: Iron titrations of UV irradiated seawater containing model ligands in competition with TAC, SA5, and SA25 as added ligand, [FeAL] versus total dissolved Fe. The same data as in Figure 1 is presented but with a log log transformation. The lines represent back calculated titration curves with the data from Table 2, the markers are the actual data points. The blue lines in 2A and D, are back calculated titration curves using theoretical $K^{cond}$ calculated from the thermodynamic constants and 2 nM as model ligand concentration. A=DTPA, B=phytic acid, C=desferrioxamine B, D=ferrichrome, E=ferrioxamine E (saturated with Fe), F=vibriobactin, G=fulvic acid FA, H=humic acid HA. Data for TAC and SA5 are from duplicate experiments, for SA25 from single experiments, except for FA and HA where for SA25 also duplicate experiments were done. Note the different HA concentrations, 0.1 and 0.2 mg.**

**DTPA**

All applications have been calibrated by reverse titration with DTPA. We would expect to recover comparable binding parameters for DTPA during the Fe titration. However, in all cases the $\log K^{cond}$ for DTPA calculated from the Fe titration was overestimated. The overestimation of $K^{cond}_{FeDTPA}$ for all three added ligands is likely a result of $\alpha_{FeDTPA} < D$ and thus theoretically below the detection window for all applications. For determinations in marine samples, Caprara et al. (2016)

showed in a compilation of data from the open ocean that with the exception of NN, the ligands were above D of the used AL, thus deviation caused by ligands with $\alpha < D$ is likely a minor problem in sweater samples. For TAC and SA5 [L] was underestimated. Although such a discrepancy in [L] could be a result of incorrect estimation of the Fe present in the titration, analysis with ICPMS showed that the Fe concentration increased only by 0.04 nM upon 2 nM DTPA addition, and thus we ruled out contamination as a cause for the underestimation of [L] for TAC and SA5. The $K^{cond}$-values are comparable between

the applications (ratios vary between 1 and 1.03, Table 3) although the range 21.3-21.8 is substantially higher than 19.01, the $K^{cond}$ used for the calibration. Due to the codependence the DTPA ligand concentration (2nM) should have been





underestimated (Apte et al., 1988), which is the case for the results from TAC and SA5 (by a factor of 0.56-0.87, Table 3). But the DTPA ligand concentration was overestimated by SA25 (by a factor of 1.31, Table 3, also see titration in Fig. 1). Indeed, in the log-log plots (Figure 2A) our results are well off from the theoretical titration line. Additionally, at the very low

concentrations the data points of all applications deviate from the modeled curves.

Phytic acid

The differences between the applications are also not large for phytic acid, the two SA applications are even quite similar. The large difference for the $K^{cond}$-values of phytic acid estimated by TAC was not expected when comparing the two very similar titration curves. We found that small changes in determined FeTAC2 concentrations at low Fe additions could be responsible

for this difference (Figure 2B). When the calculation was repeated for the combined titrations we obtained $\log K^{cond} = 22.19$ ($\pm 0.24$ and 0.15) and [L]=1.6 $\pm 0.1$.

**Siderophores**

The siderophores have high $K^{cond}$, so high that the AL should not be able to compete. However, although, the here estimated

$K^{cond}$-values are highest compared to other model A ligands, still curved titrations were obtained (Figure 1C, D, E) although there is considerable variance (. $K^{cond}$=22.1-23.5, 21.36-22.89 and 20.15-21.69 for TAC, SA5 and SA25, respectively). The $K^{cond}$, obtained for ferrichrome are close, almost identical, to those for desferrioxamine B for the three applications. However, [L] obtained for ferrichrome are higher than found for desferrioxamine B, with a factor 1.4-1.6. The $K^{cond}$-values for desferrioxamine B are lower than the value calculated from thermodynamic stability constants (Hider and Kong, 2010; Schijf

and Burns, 2016) (Tables 2 and 3). Although we want to focus on comparing the applications and not on the exact values, we need here to compare with literature values. The $K^{cond}$= 20.2 for desferrioxamine B obtained for SA25 is much lower than measured by Rue and Bruland (1995) $K^{cond} \geq 23$, although they recovered 100% of the added 2.5 nM. However, we note that they used another protocol and applied 4 minutes of nitrogen purging before every measurement which would have interfered with the signal stability according to Abualhaija and van den Berg (2014). Buck et al., (2000) also successfully recovered 100

% of a different siderophore (aerobactin) using SA25. Witter et al. (2000) measured siderophores with CLE-AdCSV, but using NN and their results compare better with the results as obtained here. They found a range of $K^{cond}$ values for a range of siderophores, and measured 21.6 for both desferrioxaine B and ferrichrome. These values are very close to those obtained here by SA5 (21.4-21.5). However, their $K^{cond}$ for phytic acid was $K^{cond}$ higher (22.3) than what we found with all the SA applications. The ratios of added [L] and obtained values by CLE-AdCSV by Witter et al. (2000) varied between 0.8 to 1.7,

resembling our results, although the ratio was 1 for both desferrioxamine B and ferrichrome. Thus, even for model ligands there is no consistence in the literature between ligands or methods suggesting problems in the standardization of the methodology.





The theoretical titration curve for desferrioxamine B has a relatively large offset at low concentrations compared to the modelled results (Figure 2C). This indicates that the theoretical $K^{cond}$ was not even approached by the three applications. That we obtained (and not for the first time) clearly curved titrations, where we should not within the applied D values, is hard to explain. One possible explanation could be a reaction taking place at the electrode surface in CSV promoting ligand exchange of Fe(III) siderophore complexes, which is producing a current. Another alternative explanation might be aluminium competition (which is not accounted for by the thermodynamic constants) since Al complexes with siderophores are detected in MS analysis of samples (Gledhill et al., 2019). The Al content however, is unknown.

Ferrioxamine E was the only model ligand that was saturated with Fe prior to the start of the experiment. Moreover, none of the ALs should sequester Fe from ferrioxamine E, which is required in order to estimate $K^{cond}$ because its $K^{cond}$ is too high, outside D (Apte et al., 1988; Hudson et al., 2003; Gerringa et al., 2014). This can be explained by considering the Langmuir isotherm, used to derive $K^{cond}$ and $[L_t]$,

$$[FeL] = \frac{K^{cond}[L_t][Fe^{3+}]}{K^{cond}[Fe^{3+}]+1},$$ (5)

Or

$$\frac{[FeL]}{[L_t]} = \frac{K^{cond}[Fe^{3+}]}{K^{cond}[Fe^{3+}]+1}$$ (6)

which shows that when $K^{cond}*[Fe^{3+}] = 1$, $[FeL] = 0.5$ [L], the equivalence point of the titration, where an almost linear relationship between [FeL] and [L] changes into an asymptotic relationship. In an Fe titration when $K^{cond}[Fe^{3+}] \gg 1$, [FeL] will approach [L]. When a titration starts at initial [FeL]>0.5[L] the ability to estimate $K^{cond}$ diminishes substantially. In the asymptote, at much larger values of [FeL]/[L], the dependence of $K^{cond}$ is lost and $K^{cond}$ becomes impossible to derive. Therefore, the titration of ferrioxamine E was more or less a standard addition and in theory [L] should be equivalent to the determined DFe concentration. However, all three applications overestimated the ligand concentration, but the estimated [L] of ferrioxamine E compares very well for TAC and SA5, and even SA25. The relatively large difference in resulting $K^{cond}$ between the two SA-applications for the hydroxamate siderophores was not expected since the D-values are close. The two most probable explanations are D and lack of equilibrium. Possibly the siderophores have $\alpha_{FeL}$ at the borders of and >D and therefore a small decrease in D still had a consequence for the outcome of the calculations. The short waiting time may be the other reason of the deviation of SA25, which we will discuss in the next section.

The [L] of the catechol Vibriobactin was underestimated by SA5 and TAC and overestimated by SA25. We believe that this divergence was a combination of lack of equilibrium due to the high stability of Fe-vibriobactin complexes during the short equilibrium period of SA25 and consequent overestimation of [L], and possibly by the tendency of catecholates to oxidise or hydrolyse in water (Brickman and McIntosh, 1992), which could have resulted in partial loss of vibriobactin during overnight equilibration.



### Humic substances

Our titrations of FA and HA show remarkable differences between the applications, [L] by TAC was 13-25 % of [L] by SA25 (Table 3). TAC detected 55 - 70% of FA and HA in contrast to an early report that TAC could not detect any portion of IHSS humic reference material (Laglera et al., 2011). Our FA and HA results are in line with the partial detection of HS by TAC that was observed already by several field studies, where an increase in $[L]_{TAC}$ correlated with an increase in natural humics (Gerringa et al., 2017; Dulaquais et al., 2018; Slagter et al., 2017,2019; Laglera et al., 2019). HS showed the largest deviations

from the expected (literature) results of all tested ligands in [L]. Humic substances are ubiquitous in seawater (Laglera et al., 2009; Whitby et al., 2020; Yamashita et al., 2020) and potentially more representative of the dominant fraction of dissolved organic matter actually present in seawater than the model A ligands tested here, since although siderophores are detected in seawater, they are typically only present at pM concentrations (Mawji et al., 2008, 2011; Velasquez et al., 2016; Boiteau et al.,2018). The different results between the three applications could explain a major part in the offset between the TAC and

SA methods in natural waters (Buck et al., 2012, 2015; Slagter et al., 2019; Ardiningsih et al., 2021). Moreover, the deviation in $K^{cond}$ obtained by TAC from the other two applications ($K_{TAC}^{cond}/K_{SA5}^{cond}$ up to 1.08 and $K_{TAC}^{cond}/K_{SA25}^{cond}$ up to 1.1) is greater than for most model A ligands (Table 3). This is also likely to be linked to the heterogeneity of humic substances, which means the detection window of each method will have a greater influence on the groups of binding sites titrated during the experiments. We cannot provide a definitive explanation for the [L] spread. TAC showed almost straight-line patterns for FA

and HA (Figure 1G, H), as in HS rich estuarine waters (Gerringa et al., 2007, Croot and Johansson 2000). This could be compatible with a fraction of HS being too strong and a fraction too weak to compete with TAC (both fractions would be at or beyond the upper and lower limits of D). There is an abundant presence of strong binding sites in HS that may not be outcompeted by TAC, since desferrioxamine B could also not outcompete all HS binding sites in Arctic Ocean samples (Laglera et al., 2019). Another possible explanation to the similar recoveries for FA and HA, despite their reported different

affinity for iron, is that TAC could form interactions with some of the binding groups of HS, cancelling their interaction with iron. In other words, the use of TAC would not obey Langmuir assumption 5. For the SA applications, [L] with SA25 seems to be substantially over the literature values in contrast to SA5. Titration data of HA with SA5, showed detectable levels of Fe-AL at low total Fe concentrations, while for SA25 they could not be seen. Thus, the formation of the electro-active Fe-SA complex does not happen until after 6-7 nM Fe has been added (0.1 mg HA, or over 10 nM for 0.2 mg HA). This is most

probably an effect of ongoing association and dissociation processes between Fe, SA and HA, i.e. a lack of equilibration. Another explanation could be a substantial decrease of the sensitivity cause by adsorption of free and complexed humics onto the surface of the electrode, shielding the electrode form interaction with Fe(AL) complexes Laglera et al., 2011, 2017). Adsorption of humics at the mercury electrode has been extensively discussed by Buffle and coauthors (Buffle and Cominoli, 1981; Cominoli et al., 1980) and the drop of sensitivity for CLE-AdCSV was discussed in Laglera et al. (2011 and 2017). The

SA25 application does not obey Langmuir assumption 1. This might also explain the large data spread in figure 2H for SA25.



**Overall**

The different results between applications and, for DTPA and desferrioxamine B, between known and observed conditional stability constants are mostly due to data in the first curved part of the titration as shown in figure 1 and illustrated when compared with the theoretical titration curves (Figures 2A,C).At this part of the titration curve peaks should in many cases be below the detection limit, thus the precision of these measurements is very low. The log log plots (Figures 2) emphasize the differences between expected and observed values. The observations seem to overestimate the FeAL at low metal additions. Possible reasons are:

1. electrochemical, for example a catalytic effect becoming more important at low concentrations and enhancing the signal or tiny peaks caused by impurities of the reagent or sitting in the methanol solvent as show in previous work with NN (Boye et al., 2001),

2. concentrations are more likely to be overestimated near the detection limit,

3. desorption from conditioned cells and electrode surfaces is more significant at low concentrations.

Further analysis is required in order to resolve these possibilities and verify that the response of FeAL is linearly related to [Fe′] or [Fe$^{3+}$], even at very low Fe concentrations (< 0.5 - 1 nM). Moreover, in order to obtain reliable estimates of [L] and $K^{cond}$) we suggest that samples should have [L′] greater than 2*DFe to ensure the titration starts at low enough [Fe′] or [Fe$^{3+}$] since it is this part of the curve that is used to the calculate $K^{cond}$ (i.e. where [FeL] ≤ 0.5[L′]; equations (5 )and (6)).

It is also possible that reactions occurred during the cathodic scan, which could also explain the deviating results of [L] for Vibriobactin, which was underestimated by 52-62% using TAC and 60-70% using SA5. Free catechols can be electro-active (Fakhari et al., 2008) and even a small contribution to the CSV peak from the ligand side of the complex would lead to a significant underestimation of the complexed fraction. A last explanation of underestimating ligand concentrations can be contamination after sampling for Fe determination by ICPMS took place.

Considering average [L] and the spread in [L] in the titrations with 2 nM of model A ligands (without considering the saturated ferrioxamine), average [L] was 1.51±0.32, 3.30±0.76 and 1.96±0.73 nM Eq Fe for TAC, SA25, SA5, respectively. There appeared to be model ligand and AL dependent variations in the estimation of [L] as also illustrated in Supplementary Figure S5. We can conclude that TAC underestimated most added model ligand concentrations, with a model ligand dependent degree of underestimation between 0.55-0.7 for HS to an average 0.8 (0.52-1.05) for the model A ligands. The application with SA5 both over and under-estimated model ligand concentrations but less than SA25 and TAC, respectively, although HS was overestimated by a factor 1.27-1.82 (assuming the number of binding sites from literature). The application with SA25 overestimated the concentrations by a factor 1.3-2.3 for model A ligands and 2.33-4.17 for HS. However, it must be remembered that [L] and $K^{cond}$ are not determined independently and unfortunately, comparison of thermodynamic constants with our results suggest that $K^{cond}$ cannot be estimated precisely. Here SA-applications result in worse estimates of $K^{cond}$ due to a lack of data at FeL<L, and the possibility that the ligands are outside D, the detection window. As far as we know, four publications describing an intercomparison exist. Three of these compared SA25 and TAC (Buck et al., 2012, 2016; Slagter et





al., 2019), one compared SA5 and TAC (Ardiningsih et al., 2021). In all publications, $[L_{SA}]$ is larger than $[L_{TAC}]$ when the data

is fitted with a one ligand model**)**. In Buck et al. (2016) it was concluded that, when using the same calculation method, comparison between the results of both applications seemed good with one exception that SA could measure a second ligand whereas TAC could not and therefore the total ligand concentration obtained with SA25 was always considerably larger (their figure 2E). This difference was attributed to an underestimation by TAC because TAC does not detect binding sites of humic substances and cannot discriminate a second ligand as well as SA25 (Buck et al., 2012, 2016; Slagter et al., 2019). Slagter et

al. (2019) sampled in the Arctic Ocean where the TransPolar Drift transports high concentrations of humic substances in the upper 50 to 80 m. Since the humic content was an important feature, TAC was compared with SA25 (Slagter et al., 2019). and the voltammetric determination of humic acids (Sukekava et al., 2018). Slagter et al. (2019) found $[L_{TAC}]/[L_{SA25}] = 0.6$. However, this ratio hardly varied with the concentration of humic substances, a strong indication that the underestimation of humics with the TAC method was not the only explanation for the difference in ligand concentration between the two methods.

Ardiningsih et al. (2021), compared TAC and SA5 in the Arctic Fram Strait and also concluded that the offset between TAC and SA5 could not only be directly ascribed to underestimation of binding sites in humic substances. Moreover, Ardiningsih et al. (2021) found a relatively constant $[L_{TAC}]/[L_{SA25}]$ on the Greenland shelf, but a variation in $[L_{TAC}]/[L_{SA25}]$ between 0.6 and 1 in Fram Strait. It was largely the inconsistencies in these studies, where humic substances were believed to potentially pay a key role in Fe speciation that lead to this study.

No intercomparison between SA5 and SA25 has been undertaken since the SA5 application was published in 2014. The question remains as to why SA5 and SA25 in this work give different results. One explanation might be disequilibrium of the SA25 application. To further study the equilibration process, we executed some kinetic experiments

### 4.3. Kinetic measurements

**4.3.1. In-cell kinetics**

In-cell kinetics were first performed with three types of samples: normal seawater, UV irradiated seawater and UV irradiated seawater containing DTPA at large concentrations (200 for TAC and 40 for both SA concentrations) (Figure 3). Further, in-cell kinetic experiments were done on a subset of the model ligands at lower concentrations (2nM or 0.2mg). For the model A ligands, DTPA was chosen since it was used as the calibrating ligand. Desferrioxamine B was chosen to represent the

hydroxamates and Vibriobactin the catecholates. Phytic acid was also included because the titration results of all applications were in agreement. FA was used to represent model B ligands, as heterogenous natural organic matter.

The in-cell kinetic measurements with high [DTPA] gave completely different results between TAC and SA (Figures 3 and 4, Metrohm stand used). For SA25, the peak decrease was high at t = 3 min (initial value) and drops sharply to the values close to the limit of detection within an hour in UV-irradiated seawater and UV+DTPA. For TAC we observed a slow increase of





the Fe(TAC)$_2$ concentration followed by an asymptotic change to a constant value, as expected for a product of a ligand

exchange reaction tending towards equilibrium.

Kinetic experiments with the low concentrations of model ligands in a volume of 10 ml was not pursued with the SA

applications, only with TAC. In these experiments, equilibrium between TAC and model ligands was reached after

approximately 6 h, as observed previously (Croot and Johansson, 2000), 4 h for most ligands and 6 h for desferrioxamine B

(Figure S6). Although the rate of Fe(TAC)$_2$ formation changed with the type of model ligand,  a steep increase where the

relative weaker ligands were added, DTPA, FA and phytic acid, against a slow and steady increase where desferrioxamine B

and vibriobactin were added, all model ligands followed the theoretical ligand exchange concentration evolution (Figure S6).

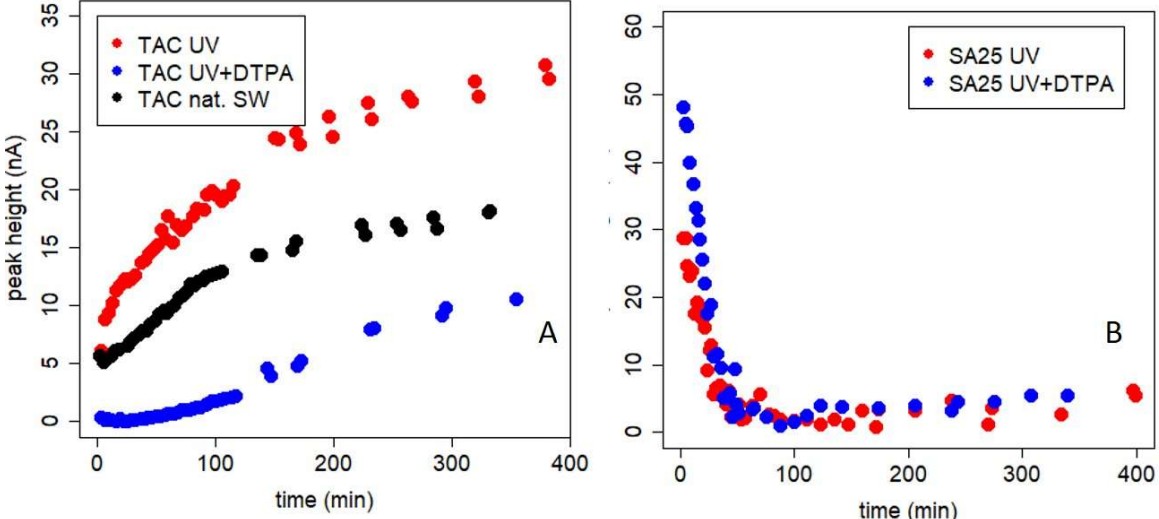

**Figure 3: In-cell kinetic experiments, with the three AL applications, in UV irradiated seawater, UV irradiated seawater + DTPA**
**(200 nM for TAC and 40 nM for the SA25 application) and natural seawater.  For SA25 UV and UV+40 nM DTPA was done. At**
**t=0 AL is added. A: TAC, B: SA25.**

Possible explanations of the rapid decrease in peak height are:

1.  The decrease can be due to formation of the non-electro-labile Fe(SA)$_2$ complex. Fe(SA)$_2$ is the non-electro-active

     species (Abualhaija and Van den Berg, 2014) and becomes the dominant Fe(SA)$_x$ complex at higher SA concentrations.

The two forms of Fe-SA would have different formation kinetics, with a slower formation of Fe(SA)$_2$, with Fe coming

     not from the dissociation of the model complex but from the dissociation of Fe(SA). This process increases the time to

     reach equilibrium and D changes accordingly. We monitored the electro-labile Fe(SA) concentrations after SA additions

     in the range 2.5 and 50 µM using the adapted Metrohm instrument with a small mercury drop (size 1) and regular air





purging (Figure 4). Possible contributions due to decreasing oxygen, and due to adsorption on mercury on the cell bottom

were thus excluded. At 25 µM SA the concentration of the electro-active species practically disappeared after 2 h. At

SA < 25 µM, the concentration of the electro-labile species Fe(SA) decreased exponentially with time for a period of at

least 13 h. At concentrations ≤ 5µM there is a decrease to a constant value above zero. These results support the formation

of a non-electro-active species Fe(SA)$_2$ irrespective of adsorption on the mercury drop, confirming Abualhaija and Van

den Berg (2014)**.**

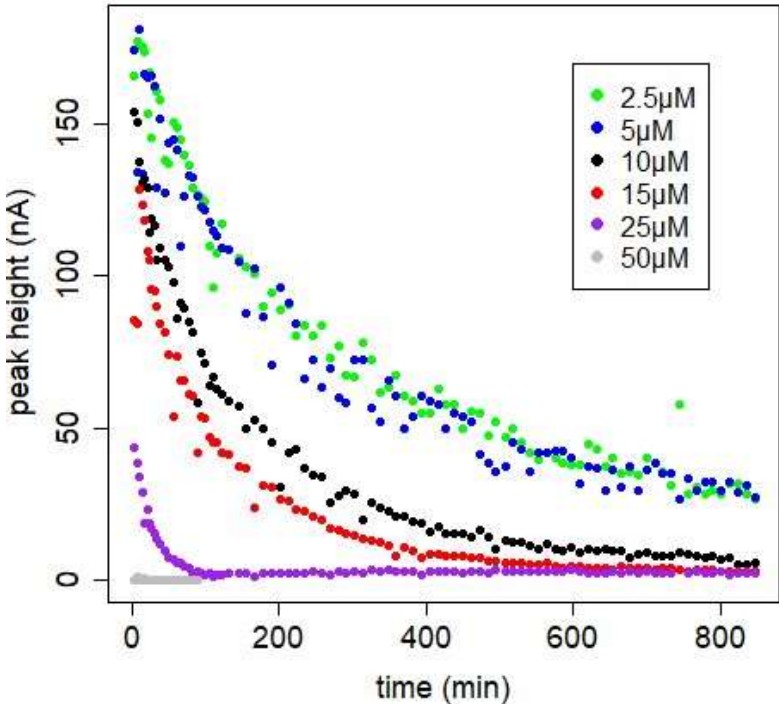


**Figure 4: In-cell kinetic experiments at different [SA], peak heights versus time. Measurements versus time are done in the same 10ml, using the Metrohm electrode; drop size =1, with regular purging with air. At t=0 SA was added**

   2.   Formation of Fe(SA)$_2$ from FeSA is slow and probably also irreversible. We investigated this possibility by trying to

force dissociation of Fe(SA)$_2$ by adding the competing model ligand DTPA during the decrease of the CSV signal in a

kinetic experiment. Addition of DTPA, did show a sudden decrease in signal with SA5, but not with higher SA

concentrations (the experiment was done at 5 and 15 µM). We suggest that at the low SA (5 µM) DTPA competed with

FeSA, causing a decrease in FeSA and thus in peak height. Adding DTPA at the higher SA concentration of 15 µM,





where Fe(SA)$_2$ is dominant, only a slight decrease in peak height was possible because only FeSA could dissociate and not Fe(SA)$_2$ within the two hours of the experiment (Figure 5). This result indicates irreversible formation of Fe(SA)$_2$

and has important implications for overnight equilibration.

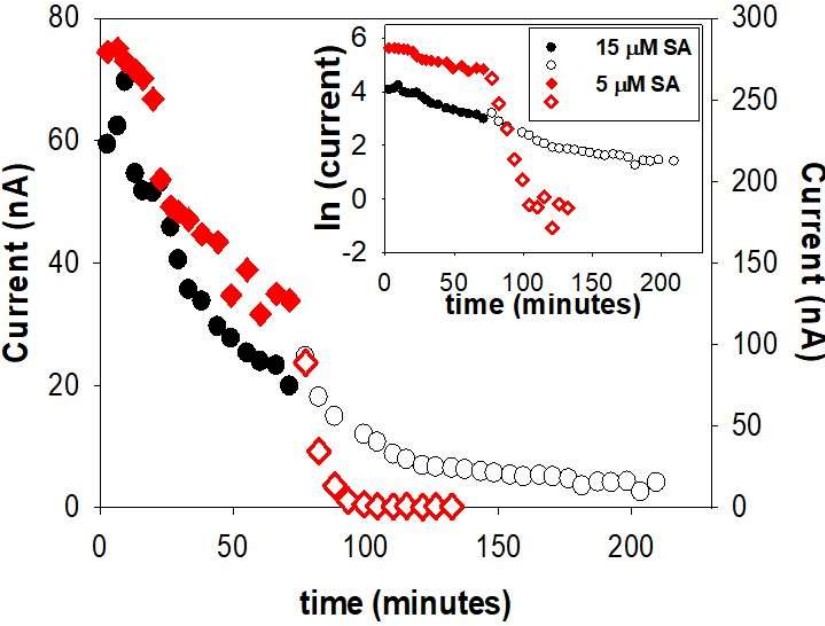

**Figure 5: Reversibility of Fe-SA formation upon addition of 150 nM DTPA at t=80 minutes. The experiment was undertaken with a Metrohm electrode in-cell in UV irradiated seawater with 6 nM Fe and two SA concentrations, 5 μM SA (right hand side Y axis) and 15 μM SA (left hand side Y axis)**

3.    Adsorption on the mercury at the bottom of the cell as indicated by Buck et al. (2007) and contradicted by Abualhaija and Van den Berg (2014). However, both used different analytical equipment, with the latter testing with a Metrohm stand, characterized by smaller mercury drops and automatic air purge. We checked the effect of the drop size for all three applications, including TAC. The TAC-application did not show any decrease in signal with time and with increasing mercury at the cell bottom. On the contrary, the two SA-applications did show a decrease that was steeper

and larger with increasing drop size. The decrease in the SA5 application was larger than in the SA25 application (Figure S3). A positive linear relationship ($R^2 = 0.98$) exists for SA25 between the decrease in peak height within 43 min and the volume of dispensed mercury at the bottom of the cell. However, no relation exists for SA5 although the reduction is strongest in that application (Figure S3). We tested whether SA was reversibly adsorbed to the mercury puddle at the bottom of the cell by transferring mercury accumulated under SA5 and SA25 protocols into a cell filled with 10 mL

UV-irradiated seawater with 6nM added Fe. No SA was released from the mercury into the seawater as no peak could





be detected when analysed with the normal AdCSV procedure. An explanation could be that only $Fe(SA)_2$ adsorbs on the mercury causing a direct relationship with peak reduction and mercury volume at high SA. Since $Fe(SA)_2$ might be formed irreversibly, no release of SA into the solution that could lead to the formation of the electroactive FeSA species would be possible. It remains hard to explain the strong reduction of the peak height without a relationship with the

mercury volume at SA5.

    4. The lack of purging influences the conditions. The BASi electrode only allows either stirring or purging in an automatic measurement, and stirring is the normal practice. Purging with air should maintain a constant concentration of oxygen and should increase the sensitivity and prevent decreasing peak heights with time (Abualhaija and Van den Berg, 2014). We checked the effect of an air purge step on the SA measurements of both electrodes, BASi and Metrohm. The decrease

in peak height with time was not influenced by an air purge step in both electrodes (Figure S2).

We can conclude that the formation kinetics of $Fe(TAC)_2$ using in-cell experiments with model ligands reached equilibrium within 8 hours. In-cell kinetic experiments with SA did not reach equilibrium and showed a continuous decay of the peak height. This can be explained by a combination of processes like adsorption of Fe(SA)-complexes on dispensed mercury at the cell bottom, and formation of the irreversible (FeSA+SA forms $FeSA_2$) and according to Abualhaija and Van den Berg (2014)

to the non electroactive $Fe(SA)_2$. The formation of irreversible species is not compatible with techniques such as CLE-AdCSV that require a dynamic equilibrium between competing ligands before analysis.

### 4.3.2 Bottle kinetics

The kinetic experiments were repeated extracting 10 mL aliquots from a 200 ml bottle. Experiments carried out were: UV-

irradiated seawater, UV-irradiated seawater with desferrioxamine B, UV-irradiated seawater with phytic acid and UV-irradiated seawater with FA (Figure 6). Some points have to be considered for interpretation. The conditioning procedure of the bottle in which the reaction takes place raises the question whether to condition with or without the AL. Addition of AL should occur at t = 0, conditioning of the bottle with AL is thus not possible. Conditioning without AL with UV-irradiated seawater with 6 nM added DFe could cause Fe precipitation and/or adsorption on the bottle walls. DFe at the end of the

experiment is probably higher than 6 nM due to Fe desorption from the bottle wall. The bottle kinetics were executed with UV-irradiated seawater, and UV-irradiated seawater with desferrioxamine B, phytic acid and FA as model ligands.







**Figure 6: Kinetic measurements for the formation of Fe-SA complexes, the change in peak height versus time. At t=0 the AL is added. In-cell means the whole experiment is done in the same 10 ml which was placed in the cell, dispensed mercury accumulates at the bottom of the cell. Bottle experiment means for every measurement a fresh 10 ml was taken from a large volume of sample. Here the reaction takes place in the large volume, and AL was added at t=0. A and B: UV irradiated sea water (UV); C and D: UV+desferrioxamine B; E and F: UV+phytic acid bottle (A, C, E) and in-cell experiments (B, D, F) are shown. G: For FA only bottle experiments were done with SA5 and SA25 as AL.**

TAC

The results with TAC showed the same pattern as for the in-cell experiments an increase that levels off to equilibrium for UV irradiated seawater, phytic acid and FA (Figure 6) characteristic of a ligand exchange reaction reaching a steady state. Higher equilibrium signals prove an extra Fe input from the bottle wall. The experiment with desferrioxamine B did not level off after 8 h and the slope was less steep. This concurs with our in-cell observation that desferrioxamine B dissociates more slowly.

That equilibrium is not reached even after 8 h explains, at least for this model ligand, the underestimation of [L] by TAC (Table 2; Croot and Johansson, 2000). It also indicates that with this protocol kinetics were not just dependent on ligand exchange but also on Fe desorption and/or redissolution of precipitated Fe.

SA

The results of the bottle experiments for SA5 changed dramatically with respect to those of the in-cell protocol. They show that equilibrium appears to be achieved for phytic acid and fulvic acid, and the increase of signal over time in the first 1-2 h is similar to that for TAC (Figure 6). An equilibrium is reached after approximately 4 h with phytic acid and FA. More time, 8 h, is needed to reach an equilibrium with desferrioxamine B, in line with the slower formation of $FeTAC_2$ in the presence of desferrioxamine B compared to the other model ligands (Figures 6, S5).

Equilibrium or a steady state is difficult to establish for SA25. A plateau is reached after approximately one hour. This seemingly depends on the added ligand, but after the plateau the signal decreases at a steady rate. We can conclude thus far that the steep decrease shown in the BASi electrode in the in-cell kinetics does not happen with the bottle experiments. This result points to adsorption on the dispensed mercury puddle as the main cause of the disappearing signal. Equilibrium is not reached at short waiting times after addition of 25 µM SA. Therefore, the use of the Langmuir isotherm to calculate the ligand

characteristics is not possible. We hypothesize that the distinction in more than one ligand group, which was often possible with this application could have been caused by the absence of equilibrium.

5. **Conclusions**

All applications have drawbacks, however the SA25 application clearly does not obey the main assumption of the Langmuir isotherm, no equilibrium is reached and therefore the results cannot be reliable. Next to this, the most upsetting conclusion is

that the estimation of the conditional stability constant $K^{cond}$ is a very rough estimate only and systematically biased by the AL. Comparing [L] obtained by the three applications with the added concentrations of the model ligands, [L] is





underestimated by TAC with a factor 0.5 to 1.05 for model A ligands and 0.52-0.7 for HS, model B ligands. The SA5 application both under- and over- estimated ligand concentrations (0.57-1.5 for model A ligands and 1.27-1.8 for model B ligands), but our kinetic studies suggest that true equilibrium may still be an issue for strong ligands. The SA25 application

overestimated [L] by a factor 1.31-2.3 for model A ligands and 2.33-4.17 for model B ligands. Moreover, for all approaches, ligand specific interferences occurred, as for humic acids.

We confirm the conclusion of Abualhaija and van den Berg (2014) that the SA concentration needs to be low ≤5μM to prevent formation of not electro-active $Fe(SA)_2$. Our experiments suggest that the formation of $Fe(SA)_2$ is irreversible, and thus does not obey the Langmuir equation.

Probably $Fe(SA)_2$ also adsorbs on dispensed mercury on the bottom of the cell. When using a BASi electrode small mercury drops are to be preferred.

It is time to search for other methods to study the overall effect of organic ligands on iron speciation in marine waters. However, in case voltammetric methods are used, we advise the SA5-application but to be careful to draw conclusions from obtained $K^{cond}$-values, since apart of the constrained forced by D, these estimates have a much larger error than expected by fitting a

Langmuir isotherm.

It is clear that further work needs to be done to effectively contextualise the data base and robust quality control procedures are urgently required. We recommend that these procedures include the determination of a standard ligand with independently determined thermodynamic constants that is within the detection window of the applied methodology. Our work suggests that none of the ligands examined here are ideal for this, since they might have been outside the detection window, or do not have

available thermodynamic constants for comparison.

### Acknowledgements

We are grateful for the help of Kristin Buck during the time we struggled to get the SA method working. The bachelor students David Amptmeijer, Robert Sluijter did preliminary experiments whereas Ismael Salazar and Martijn Korporaal executed experiments used in this manuscript. The critical reading of Rebecca Zitoun helped to improve the manuscript.

The PhD work of Hans Slagter and co-author Indah Ardiningsih made this research inescapable. IA is funded by Indonesia endowment fund for education (LPDP). LML received support from the MICINN project CTM2017-84763-C3-3-R.





**Tables and Table captions**


Table 1: Average beta and alpha values of the added ligands (AL) with the standard deviation around the mean of N experiments. In bold the parameters used in this study to calculate the model ligand characteristics, 1: assuming one FeAL is formed, either FeSA or Fe(SA)$_2$, 2: assuming both FeSA and Fe(SA)$_2$ are formed. a-c indicate literature values: [a] Croot and Johanson (2000) using αinorg =10; [b] Abualhaija et al. (2015) using αinorg =9.98; [c] Buck et al. (2007) using αinorg =10.

Alpha values of the AL are the direct outcomes of the calibration exercises; therefore, these have a standard deviation added, which is the standard deviation around the mean of 4 calibrations. Since K and/or β are directly derived by dividing trough the AL concentration or squared concentration the standard deviations of $logK^{cond}_{FeAL,Fe'}$ and $log\beta^{cond}_{FeAL\ ,Fe'}$ have the same values.

| AL | $\alpha_{FeAL,Fe'}$ | $log\alpha_{FeAL,Fe3+}$ | N | $log\beta^{cond}_{FeAL2,Fe'}$ | $log\beta^{cond}_{FeAL2,Fe3+}$ | or | $logK^{cond}_{FeAL,Fe}$ | $logK^{cond}_{FeAL,F}$ |
|---|---|---|---|---|---|---|---|---|
| TAC | **275** 250[a] | **12.3±0.2** 12.4[a] | 4 | **12.4** 12.4[a] | 22.3 22.4[a] | | | |
| SA5 | **4.16** 17.87[b] | **11.1±0.1** 11.23[b] | 4 | 11.3[1] 10.7[b] | 21.7[1] 20.7[b] | | **5.9**[2] 6.5[b] | **16.3**[2] 16.5 [b] |
| SA25 | **5.49** 78.7 [c] | **11.3±0.23** 11.9 [c] | 5 | 10.1[1] 11.1 [c] | 20.5[1] 21.1 [c] | | **5.34**[2] | **15.8**[2] |

Table 2: Results of the titrations following the three applications, SA5, SA25 and TAC for model ligands A, with a well described composition, and a specific added concentration of 2 or 4 nM; and for model ligand B, the humic substances FA and HA, that do not have a fixed composition and were added in weight units (01-0.4 mg). DFe was measured by ICPMS, $logK^{cond}$ and [L] are calculated using the non-linear Langmuir isotherm. Alpha ($K^{cond}*[L']$) is calculated calculating [L'] and not by simple [L]-DFe. DFe in nM, [L] in nM Eq Fe, $K^{cond}$ in M$^{-1}$. For the model ligands 2 nM were used unless otherwise stated.

Most model ligands have been analysed in duplicate with TAC and SA5, and once with SA25. The addition of the humics was determined using Laglera and Van den Berg (2009) for HA and Sukekava et al., (2018) for FA.

Since K' is log transformed the standard error (SE) is asymmetric to lower and to upper values, therefore two SE values are obtained, one to lower (down) and to upper (up) values.

NA: SE Down could not be determined for data that fitted the Langmuir isotherm less good.

[1] Unreliable result, Fe is added up to 12.5 nM, therefore [L]=13.23 cannot be calculated in a correct way, even though the SD of the fitted value is relatively low.





| name | TAC |  |  |  |  |  |  | SA 5µM |  |  |  |  |  |  | SA 25µM |  |  |  |  |  |  |
|---|---|---|---|---|---|---|---|---|---|---|---|---|---|---|---|---|---|---|---|---|---|
|  | DFe | logK | SE | SE | [L] | SE | log | DFe | logK | SE | SE | [L] | SE | log | DFe | logK | SE | SE | [L] | SE | log |
|  | nM |  | Down | Up | nM |  | $\alpha_{Fe'}$ |  |  | Down | Up | nM |  | $\alpha_{Fe'}$ |  |  | Down | Up | nM |  | $\alpha_{Fe'}$ |
| Model A |  |  |  |  |  |  |  |  |  |  |  |  |  |  |  |  |  |  |  |  |  |
| DTPA | 0.31 | 21.8 | 0.44 | 0.21 | 1.35 | 0.20 | 2.96 | 0.12 | 21.68 | NA | 0.44 | 1.15 | 0.09 | 2.29 | 0.25 | 21.25 | 0.26 | 0.16 | 2.62 | 0.15 | 2.23 |
| 2nM | 0.31 | 21.7 | 0.24 | 0.16 | 1.73 | 0.24 | 2.90 | 0.12 | 21.55 | NA | 0.56 | 1.14 | 0.21 | 2.16 |  |  |  |  |  |  |  |
| PhA | 0.29 | 21.9 | 0.15 | 0.11 | 1.85 | 0.13 | 3.18 | 0.44 | 20.71 | 0.25 | 0.16 | 2.85 | 0.30 | 1.69 | 0.57 | 20.13 | 0.27 | 0.17 | 3.12 | 0.95 | 1.14 |
| 2nM | 0.29 | 22.8 | NA | 0.34 | 1.32 | 0.09 | 2.92 | 0.44 | 20.79 | 0.27 | 0.17 | 2.34 | 0.20 | 1.67 |  |  |  |  |  |  |  |
| DesferB | 0.29 | 22.4 | 0.58 | 0.24 | 1.43 | 0.12 | 3.53 | 0.3 | 21.44 | 2.36 | 0.30 | 1.80 | 0.13 | 2.22 | 0.43 | 20.15 | 0.25 | 0.16 | 2.98 | 0.84 | 1.16 |
| 2nM | 0.29 | 22.3 | 0.43 | 0.21 | 1.41 | 0.11 | 3.47 | 0.3 | 21.44 | NA | 0.38 | 2.04 | 0.20 | 2.28 |  |  |  |  |  |  |  |
| Ferchr | 0.61 | 22.1 | 0.16 | 0.12 | 2.1 | 0.11 | 3.40 | 0.42 | 21.46 | 0.62 | 0.24 | 2.66 | 0.16 | 2.41 | 0.55 | 20.27 | 0.24 | 0.15 | 4.6 | 0.99 | 1.48 |
| 2nM | 0.61 | 22.6 | 0.19 | 0.13 | 1.66 | 0.05 | 3.68 | 0.42 | 21.36 | 0.37 | 0.20 | 3.00 | 0.15 | 2.37 |  |  |  |  |  |  |  |
| Ferriox | 2.39 | 22.7 | 0.27 | 0.16 | 2.79 | 0.08 | 3.35 | 2.2 | 22.51 | NA | 0.79 | 2.84 | 0.16 | 2.92 | 2.33 | 21.69 | 0.24 | 0.15 | 3.94 | 0.12 | 2.50 |
| 2nM | 2.39 | 22.8 | 0.53 | 0.23 | 2.77 | 0.08 | 3.50 | 2.2 | 22.89 | NA | 1.08 | 2.83 | 0.16 | 3.29 |  |  |  |  |  |  |  |
| Ferriox | 4.34 | 23.1 | 0.23 | 0.15 | 5.03 | 0.06 | 4.04 | 4.15 | 22.32 | NA | 0.58 | 5.01 | 0.14 | 2.86 | 4.28 | 21.63 | 0.19 | 0.13 | 6.81 | 0.14 | 2.63 |
| 4nM |  |  |  |  |  |  |  | 4.15 | 22.17 | 0.79 | 0.26 | 5.26 | 0.08 | 2.82 |  |  |  |  |  |  |  |
| Vibrio | 0.58 | 23 | NA | 0.38 | 1.25 | 0.07 | 3.94 | 0.39 | 21.62 | NA | 0.54 | 1.40 | 0.19 | 2.23 | 0.52 | 21.21 | 0.14 | 0.11 | 3.18 | 0.18 | 2.24 |
| 2nM | 0.58 | 23.5 | NA | 0.51 | 1.03 | 0.05 | 4.21 | 0.39 | 21.48 | NA | 0.50 | 1.19 | 0.16 | 1.99 |  |  |  |  |  |  |  |
| Model B |  |  |  |  |  |  |  |  |  |  |  |  |  |  |  |  |  |  |  |  |  |
| FA, 0.2 | 1.39 | 22.3 | 0.18 | 0.13 | 2.04 | 0.08 | 3.21 | 1.2 | 20.62 | 0.21 | 0.14 | 3.68 | 0.32 | 1.62 | 1.33 | 20.41 | 0.13 | 0.10 | 8.11 | 1.28 | 1.84 |
| mg/l | 1.39 | 22.6 | 0.81 | 0.27 | 1.69 | 0.11 | 3.13 | 1.2 | 21.22 | 0.22 | 0.15 | 3.95 | 0.15 | 2.26 | 1.33 | 20.58 | 0.10 | 0.08 | 6.75 | 0.65 | 1.92 |
| HA, 0.1 |  |  |  |  |  |  |  | 2.2 | 20.69 | 0.30 | 0.18 | 5.43 | 0.47 | 1.80 | 2.33 | 20.57 | 0.05 | 0.04 | 10.45 | 1.01 | 2.08 |
| mg/l |  |  |  |  |  |  |  | 2.2 | 20.79 | 0.16 | 0.12 | 5.82 | 0.35 | 1.95 |  |  |  |  |  |  |  |
| HA, 0.2 | 3.89 | 21.7 | 0.23 | 0.15 | 3.69 | 0.28 | 1.35 |  |  |  |  |  |  |  | 2.33 | 21.16 | 0.09 | 0.07 | 13.33[1] | 0.68 | 2.80 |
| mg/l | 3.89 | 21.9 | NA | 0.37 | 3.49 | 0.75 | 0.98 |  |  |  |  |  |  |  |  |  |  |  |  |  |  |




Table 3: The differences between the results in Table 3 of the three applications, ratio's or the difference in concentration are given. Added model ligand concentrations are given in column 1. The far right column contains the percentual deviation from the added ligand concentration as $E(\%) = (([L_{AL}]-[])/[])\times100$, with $[L_{AL}]$ as the result of the

applied ligand method and $[]$ as the added concentration of the model ligand.

Data containing the ligand concentration, are from the model ligands added at a concentration of 2 nM and thus excludes HA and FA. For these we used the ligand site concentrations of 2.92 and 6.4 nM Eq Fe for 0.2 mg added fulvic and humic acids from Sukekava et al. (2018) and Laglera and Van den Berg (2009). Since humic acids are not discrete ligands the estimate %E are in italic.




| | logarithm values | | | | | | | | | $E_{AL}$(%) = ([$L_{AL}$]-[])/[])x100 % | | |
| | Log values | | | Log values with respect to $Fe^{3+}$ | | | | | | | | |
| | $K_{SA5}$/ $K_{SA25}$ | $K_{TAC}$/ $K_{SA5}$ | $K_{TAC}$/ $K_{SA25}$ | $\alpha_{SA5}$/ $\alpha_{SA25}$ | $\alpha_{TAC}$/ $\alpha_{SA5}$ | $\alpha_{TAC}$/ $\alpha_{SA25}$ | $L_{SA5}$/ $L_{SA25}$ | $L_{TAC}$/ $L_{SA5}$ | $L_{TAC}$/ $L_{SA25}$ | | | |
| Name | | | | | | | | | | $E_{SA5}$ | $E_{SA25}$ | $E_{TAC}$ |
| **Model A** | | | | | | | | | | | | |
| **DTPA** | 1.02 | 1.01 | 1.03 | 1.01 | 1.01 | 1.02 | 0.44 | 1.18 | 0.52 | -43 | 31 | -33 |
| **2nM** | | 1.00 | | | 1.02 | | | 1.52 | | -43 | | -14 |
| **Phytic acid** | 1.03 | 1.06 | 1.09 | 1.05 | 1.08 | 1.13 | 0.91 | 0.65 | 0.59 | 42 | 56 | -8 |
| **2nM** | | 1.10 | | | 1.06 | | | 0.56 | | 17 | | -35 |
| **Desferrioxam. B** | 1.06 | 1.04 | 1.11 | 1.09 | 1.06 | 1.16 | 0.61 | 0.79 | 0.48 | -10 | 49 | -29 |
| **2nM** | | 1.04 | | | 1.05 | | | 0.69 | | 2 | | -30 |
| **Ferrichrome** | 1.06 | 1.03 | 1.09 | 1.08 | 1.04 | 1.12 | 0.58 | 0.79 | 0.46 | 33 | 130 | 5 |
| **2nM** | | 1.06 | | | 1.06 | | | 0.55 | | 50 | | -17 |
| **Ferrioxamine E** | 1.04 | 1.01 | 1.04 | 1.03 | 1.00 | 1.03 | 0.72 | 0.98 | 0.71 | 42 | 97 | 39 |
| **2nM** | | 1.00 | | | 0.98 | | | 0.98 | | 42 | | 38 |
| **Ferroxamine E** | 1.03 | 1.03 | 1.07 | 1.02 | 1.05 | 1.07 | 0.74 | 1.00 | 0.74 | 25 | 70 | 25 |
| **4nM** | | | | | | | | | | 31 | | |
| **Vibriobactin** | 1.02 | 1.06 | 1.08 | 1.00 | 1.10 | 1.10 | 0.44 | 0.89 | 0.39 | -30 | 59 | -38 |
| **2nM** | | 1.09 | | | 1.14 | | | 0.87 | | -40 | | -49 |
| **Model B** | | | | | | | | | | | | |
| **FA** | 1.01 | 1.08 | 1.09 | 0.98 | 1.09 | 1.07 | 0.45 | 0.55 | *0.25* | *26* | *178* | *-30* |
| **0.2mg/2.9nM** | 1.03 | 1.06 | 1.10 | 1.03 | 1.03 | 1.06 | 0.59 | 0.43 | *0.25* | *35* | *131* | *-42* |
| **HA** | 1.01 | 1.05 | 1.06 | 0.98 | | | 0.52 | 0.34 | *0.18* | *70* | *226* | *-41* |
| **0.1mg /3.2nM** | 0.98 | 1.05 | 1.03 | 0.94 | | | 0.44 | 0.30 | *0.13* | *82* | *317* | |
| **HA** | | | | | | | | | | | | *-43* |
| **0.2mg /6.4nM** | | | | | | | | | | | | *-46* |





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
