# Peer review of "Comparing CLE-AdCSV applications using SA and TAC to determine the Fe binding characteristics of model ligands in seawater"

_Biogeosciences, 2021_

## Community Comment (CC2)

**Comments on "Comparing CLE-AdCSV applications using SA and TAC to determine the Fe binding characteristics of model ligands in seawater"**

The authors present a very careful and interesting study that compares some of the most common CLE-AdCSV techniques used to measure the organic speciation of dissolved iron (dFe) in seawater. These experiments are very painstaking to do, so I commend the authors on such an undertaking. Overall, I found the comparison between techniques using model ligands an important contribution to the field, but unfortunately it appears that there might have been some major issues with the application of the SA methods particularly in the kinetic experiments (e.g. Figure 6). On a broader note, I also believe that this paper is demanding more than can be expected out of these electrochemical methods. These voltammetric approaches have always been operationally defined (as all seawater iron analysis methods are) and it has been known for some time that applying these methods at different analytical windows will give you a method-specific perspective on the continuum of iron ligands in seawater. Bruland et al. (2000) highlighted this very eloquently, and although that intercomparison applied to Cu speciation, its findings are applicable to Fe speciation as well. The main goal of these CLE-AdCSV methods has never been to quantify a specific ligand's concentration and binding strength, but instead to get a broad picture of the multitude of ligands that are capable of binding Fe in seawater, and to qualitatively evaluate this in seawater where a continuum of ligands exists. From studies where these methods have been applied on the basin scale, we have gained unparalleled insights into Fe and ligands dynamics in the ocean, that are robust and oceanographically consistent and are able to be captured in global biogeochemical models (Tagliabue et al. 2017). Thus far, these are the only methods we currently have to learn about the Fe-binding ligand pool as a whole. Other methods exist to measure specific ligands and functional groups, and what we have learned from those techniques is thus far unable to help us explain global distributions of dFe. It would be a shame in my opinion, to not comment on the valuable insights CLE-AdCSV techniques have brought us over the years, and to reduce the findings of this paper to a take home message that these methods are fatally flawed and we need new methods for measuring organic Fe speciation. Innovation in techniques is always a good idea, but I worry about how this paper might impact the direction of the field. Given the stature and expertise of the authors, I would have appreciated some discussion of the strengths of CLE-AdCSV methods and the instances of where and when they do work, and where and when they should be applied with caution and pause. I have outlined some of my other major comments and thoughts below, and have noted some additional specific comments at the end of the document.

**General comments**

Kinetic experiments and the loss of signal with the SA method: The authors mention that all vials were conditioned overnight prior to the start of experiments, but in my lab, we have found that thorough conditioning for usually more than one week is absolutely required for accurate titrations and optimal sensitivity with the SA method (Abualhaija and van den Berg, 2014). We are not sure why this is the case with SA, but we have observed it repeatedly as we purchase new Teflon vials and condition them. As most titrations that we perform in the lab take several hours to complete and no SA signal loss is observed over that time frame, the dramatic loss of SA signal in the kinetic experiments after only minutes in some cases, is alarming. We also routinely perform overnight equilibrations with SA at 5, 10 and 25 µM in my lab with no loss of signal. Was the effect of conditioning ever tested with SA in this study? It is possible that the formation

of a non-electroactive $Fe(SA)_2$ complex at higher SA concentrations (and also higher Fe concentrations) is related to this conditioning issue. For example, we have found that adding high Fe concentrations to our Teflon vials along with buffer and the appropriate SA concentration yields the best conditioning results, and no resultant loss in signal. Perhaps this might be because $Fe(SA)_2$ is formed under the conditions of the conditioning, and this complex has different adsorption properties to Teflon than the Fe(SA) species. The ultimate reason for the conditioning is unclear to me, however it is clear that with proper conditioning no loss of SA signal should be observed over time in the vials. I am worried that the results of the kinetic studies and even the model ligand results are over shadowed by these potential conditioning issues, rather than a lack of equilibrium of Fe with SA.

The authors discuss the impact of mercury drops in the titration cell on SA measurements and first say that the mercury should have no effect (since they did this experiment also on a Metrohm with smaller mercury drops) and that the drop in signal is instead due to a disequilibrium because of the formation of $Fe(SA)_2$. They then seem to contradict this idea later in the manuscript. Accumulation of mercury in the titration cell is a big problem for SA measurements, because the adsorption potential for Fe(SA) is 0V. Thus, uncharged mercury in the bottom of the cell can adsorb mercury and thus competes with the "active" drop for the binding of Fe(SA). The other issue is that the accumulation of mercury in the bottom of a cell, for a BASi instrument in particular (where a stir bar is used instead of a suspended rod), can also physically impede the stirring process, which then dramatically decreases the sensitivity (peak height) of the measurement. The bottle versus in-cell kinetic experiments perfectly illustrates this, and yet the discussion of these results is largely presented in the context of the disequilibrium argument for SA. I think the authors should make this result very clear in the manuscript, and highlight that this observation is a not a definitive support of disequilibrium. I also do not really understand the goal of the experiments where the mercury "puddle" is placed into fresh seawater and then no additional Fe(SA) signal is observed. I may have misunderstood how this experiment was performed, but based on how it is written in the text I do not think a lack of signal in this experiment signifies irreversible formation of Fe(SA), but instead reflects the fact that Fe(SA) adsorbs at 0V and is not reversibly removed until a negative potential is applied to the mercury drop(s).

Comparison of the TAC and SA methods with model ligands: I found this section very interesting, and it is an important aspect of this work to report to the community. However, the model ligand section is hard to follow because it is not always clear what the measured values are being compared to in terms of the expected or "true" value. The expected values of each model ligand based on what has been seen previously in the literature would be immensely helpful to include in Table 2. Also, please label each $logK^{cond}$ with either a $Fe^{3+}$ or Fe' subscript, because I was often getting confused which one you were reporting in some figures, tables and in the text, particularly in Table 2. For example, you report the model A ligand $logK^{cond}$ with respect to $Fe^{3+}$ and the model B ligands with respect to Fe'. Yet, in Table 2 you report the results for model B ligands with respect to $Fe^{3+}$. Please keep these consistent so that comparison across methods and to previous studies are possible. I also think it would be helpful to report the "true" or expected values for each model ligand because that might also make Table 3 more meaningful. Table 3 is useful in terms of how the different methods perform relative to one another, but how about how they perform relative to what is the expected value? This would be

much more insightful to consider. I actually had to make my own table while I was reading the manuscript in order to see for myself how the measured values compare to previous literature results (and thus the "expected" value for each model ligand). I think it is also important to thoroughly discuss the detection window being used in each method, and how that compares to the analytical window of each model ligand experiment. A quick calculation of the analytical window for each model ligand case shows that several of the titrations are likely outside of the analytical window for some of the methods. For example, based on the expected logK for each model ligand from previous literature values, the TAC method was best suited for measuring most of the model siderophores and it often performed better than SA for these model ligands. In the opposite case, the SA 5 µM method performed better with the weaker humic and fulvic ligands. Making the connection between analytical window and the model ligand being examined clear in light of the results obtained is critical.

Recommendations for future work and insights from past work: Given the implications of this work to the field and the extensive knowledge and background of the authors, I was hoping there would be a final section of the manuscript with recommendations going forward. The authors mention that we need to find new ways to measure the speciation of Fe in seawater, but make no suggestions. The authors should also comment on how past results might be interpreted. When I made my own table where I compared the measured ligand concentrations and logKs from each model ligand study to past results seen in the literature, to my eyes there was no systematic "best" method. It was often dependent on the model ligand and the analytical window where that model ligand falls, relative to the analytical window of the method used. Some discussion that brings all of the insights from this paper together and gives recommendations going forward beyond, "we need a better way" would be very powerful. As both an electrochemist and a mass spectrometerist, I can say with certainty that no method is perfect, and each has its benefits and pitfalls.

**Specific comments**:
There are several small typos, only some of which I have detailed here.

Section 2: You list your assumptions and refer to them by number, but they are not numbered. Numbering them might be helpful, since this whole section reads like a list and you refer to specific assumptions later in the manuscript.

Line 104: Add a space between "knowledge" and "In"

Line 108: Do you mean 20 minutes or 15 minutes? You use 15 minutes throughout the manuscript, and say that you also used a timer.

Line 129: Add a space between "[L]" and "and"

Line 181: Were the samples filtered prior to being stored frozen? If so, how were they filtered? Where they frozen at -20C?

Line 182: Was the UVSW aged prior to use or used immediately?

Line 215: Was all kinetic conditioning done in UVSW?

Line 225: Remove the second "in" after "placed"

Line 227: Do you have a reference for this?

Line 260: Bundy et al. (2018) determine the conditional stability constant of ferrioxamine E in seawater (as well as ferrioxamine B). The $\log K_{FeL,Fe'}^{cond}$ for ferrioxamine E was 14.05 and the $\alpha_{Fe'}$ used was 10. These measurements were performed using the SA method at 5 μM.

Line 308: Were blanks also in absence of Fe?

Line 313: This sentence is confusing. I think you mean that you added buffer and dFe and not also SA, and then you added SA after equilibration. Why equilibrate the buffer and dFe for one hour? The commonly used method equilibrates the buffer and dFe for two hours before adding the SA (Buck et al. 2007).

Line 321: I think you mean "prior to the addition of TAC or SA."

Line 357: Why was this dFe concentration chosen? Was your seawater a deep sample? Why not used the measured dFe concentration?

Line 369-370: In most SA CSV studies the blank is zero, meaning no Fe is added with the buffer or SA addition. Can you include the ordering information for the boric acid and SA that were used? Was distilled or Optima ammonium hydroxide used for the buffer preparation? The fact that there was a blank with these measurements is very disconcerting. On a related note, please note the error associated with the dFe measurements for the model ligand results. In some cases, the standard deviation on the concentration of L and the logK is relatively small, and given that the dFe in each experiment varied quite widely, it would be nice to see the error on these measurements as well.

Line 410: Change to "ligands like siderophores point to biases"

Line 588: I think it is difficult to make too many assumptions on the HS and FA results, because there is large variability in the literature with respect to the available dFe binding sites. Again, what is the approximate "true" value we can compare these model ligan titrations to? Did you determine your own dFe binding capacity measurements for the batches of FA and HS you used?

Line 645: You can still have mercury adsorption on the drop even with smaller drop sizes. The drop on the Metrohm itself is smaller, therefore having the same sized drops in the bottom of the cell, relative to the active drop, will still give you the same issue.

Table 1: Some of this table is cutoff, so it is difficult to understand what is being displayed.

Randie Bundy

**References**

Abualhaija, M.M. and van den Berg, C.M., 2014. Chemical speciation of iron in seawater using catalytic cathodic stripping voltammetry with ligand competition against salicylaldoxime. *Marine Chemistry*, *164*, pp.60-74.

Bruland, K.W., Rue, E.L., Donat, J.R., Skrabal, S.A. and Moffett, J.W., 2000. Intercomparison of voltammetric techniques to determine the chemical speciation of dissolved copper in a coastal seawater sample. *Analytica Chimica Acta*, *405*(1-2), pp.99-113.

Buck, K.N., Lohan, M.C., Berger, C.J. and Bruland, K.W., 2007. Dissolved iron speciation in two distinct river plumes and an estuary: Implications for riverine iron supply. *Limnology and Oceanography*, *52*(2), pp.843-855.

Bundy, R.M., Boiteau, R.M., McLean, C., Turk-Kubo, K.A., McIlvin, M.R., Saito, M.A., Van Mooy, B.A. and Repeta, D.J., 2018. Distinct siderophores contribute to iron cycling in the mesopelagic at station ALOHA. *Frontiers in Marine Science*, *5*, p.61.

Tagliabue, A., Bowie, A.R., Boyd, P.W., Buck, K.N., Johnson, K.S. and Saito, M.A., 2017. The integral role of iron in ocean biogeochemistry. *Nature*, *543*(7643), pp.51-59.

---

## Author Response (AR1)

**Reviewer Dario Omanović.**

In this work, the three commonly used CLE-AdCSV methods for speciation of Fe in marine waters were compared. Discrete synthetic ligands of known concentrations and isolated organic matter representing the natural heterogeneous ligand groups (humics) were studied. The study summarizes the experiments that confirmed the existing methodological problems of CLE-AdSCV in Fe speciation, most of which have been reported in the literature. The most important aftereffect is that this study in some way challenges the reliability of all previous research studies on Fe speciation in marine waters. This could have major consequences. The last sentence of the abstract best reflects the outcome of the manuscript:"...we need to search for new ways to determine the organic complexation of Fe in seawater".

This is my second review of this manuscript (originally submitted to Frontiers in Marine Sciences, as also noted by the authors). Most of my comments on the original version of the manuscript have been incorporated into the revised version, and I agreed that it was suitable for publication. This version is further polished, and I have no additional comments. My original review, with the authors' responses, is available upon request if it complies with journal policy.

[Figure]

Dear Dario Omanović. Thank you for your comment, thank you for your earlier elaborate comments, which improved the manuscript considerably.

**Anonymous Reviewer 1**

- General comment

**Comment:** In this manuscript, three CLE-ACSV methods (TAC, SA5, SA25) for analyzing the concentration and conditional stability constants of organic Fe-binding ligands using several model ligands (DTPA, phytic acid, desferrioxamine B, ferrichrome, ferrioxamine E, vibriobactin, FA and HA). It is notable that not only comparing the analysis results from the CLE-ACSV titrations, but also this manuscript investigates and discusses the differences in the characteristics of the competing ligands SA and TAC and differences in analytical instruments. Although these points have been pointed out as possibilities for some time, there are few studies that discuss their effects based on the actual measured value. In the Conclusion, it is suggested that future studies of organic Fe-binding ligands in seawater requires alternatives to the current CLE-ACSV method, and I agree. Although the model ligands used in this manuscript are just "model" and can be considered different from the natural ligands in the ocean, understanding the characteristics of each CLE-ACSV method was exactly what was needed for research in this area. In addition to this, I am interested in the influence of the results on the multiple analytical windows method. In recent years, a multiple analytical windows technique with SA has been applied to determine the multiple classes of Fe-binding ligands in seawater (e.g., Bundy et al., 2014, which has been refereed in this manuscript). How do you think about the influence of these results on the evaluation of multiple analytical window method?

**Answer:** We thank the reviewer for her/his comments

The question about the multiple window approach is very interesting. We think the multiple analytical window approach is a very good step forward to obtain more details on metal speciation, but since it estimates more parameters one has to consider carefully the degrees of freedom in the estimation.

First, this approach will only work if distinct groups of organic metal binding ligands do exist and are not shielded by other organic ligands which have a continuum of ligand sites (thinking of humics and the work of Buffle). Continuing in this line of thinking, this approach has more chance of success with Cu because the difference in alpha factor for Cu-binding organic ligands is larger and concentrations are larger, making distinguishing more than one ligand group easier.

Second, by applying the multiple window approach the added ligand concentration [AL] varies. From the work of Abualhaija and also from our experiments we know that probably the formed metal-added ligand ($Me_x$-$AL_y$) species varies with the [AL]. Thus, the range of formed species and whether they are yes of no electro active must be known, together with the conditional binding strengths per $Me_x$-$AL_y$ species. It surely complicates matters. For example with SA, with increasing [AL], the non-electro-active species will increase (according to Abualhaija), it might even be at cost of the electro-active species, resulting in a decrease instead of the expected increase of the signal.

Third, the kinetic problem we discuss in our manuscript in the 25µMSA application will interfere with the multiple window approach. When a short equilibration is used the slow kinetics of the

exchange between iron and natural organic ligands will probably be influenced differently at different [AL]. We think that overnight equilibration will largely overcome this problem.

Fourth, we suggest the possibility that formation of $FeSA_2$ is irreversible. We cannot prove this with the experiments we did. However, suppose we are right then this would interfere with the multiple window approach.

Comment: Overall, this manuscript is organized and well-written. I believe that this manuscript is going to have a strong influence on the development of research in this area. After responding to the following minor comments, I think this manuscript can be accepted.

**Answer:** We thank the reviewer for the interesting comment above and the suggested improvements below.

**Minor comments**

**Comment:** Line 17. Fe3+ should be $Fe^{3+}$

**Answer:** Thank you

**Comment:** Line 125~160. The "Langmuir isotherm assumption" is often mentioned in this manuscript and the outline of the assumption itself is explained. However, as shown in Line 163(…assumption 2 or 6), for example, it is not clear from this manuscript alone what each assumption number refers to, so please indicate it in the text after Line 132.

**Answer:** Indeed, this is awkward, we added 1-6 in lines 125-160.

**Comment:** Line 185~. Please indicate the temperature conditions of the samples during equilibration period for each experiment in the main text.

**Answer:** We added at line 207: "Equilibration between the samples and AL was attained at room temperature."

**Comment:** Line 227. Please show the reference information of the UV irradiation for the past experiments related to Co and Cu ligand analyses.

**Answer:** Rapp, et al. (2017) and Wuttig, et al. (2019) assessed the influence of quartz and FEP vessels on UV-digestion efficiency and contamination. No difference between a FEP bottle and a quartz cuvette was observed with regards to the efficiency of UV-digestion using either vessel material.

We added in the text the references: Rapp et al., 2017; Wuttig et al., 2019.

**Comment:** Line 331~. The explanations of the In-cell experiments and bottle experiments are complicated to understand for people outside the field, so I thought it would be nice to have a schematic diagram as a supplementary figure.

**Answer:** We understand that it is somewhat complicated, we added a scheme in the Supplementary information as Figure S4

**Comment:** Line 385. "Using Eqs. (1) and (3) give"

**Answer:** Indeed, thank you, we changed the text as suggested.

**Comment:** Figure 1. Please add [A] – [H] in each figure.

In figure [H], the HA concentration is 1 mg and 2 mg in the legend, so please correct it.

**Answer:** Apologies, this was the wrong figure, we replaced it with the correct one. Thank you for spotting this. Moreover, the complete figure is new, since we changed the colors (see comment at figure 2).

**Comment:** Line 434. Since the standard deviation can be shown with 3 or more data. If the number of data is 2, strictly speaking, the difference from the average value is the correct notation.

**Answer:** Thank you, we changed the text as suggested.

**Comment:** Figure 2. I sometimes suffered from distinguishing the color patterns in these figures. Could you change the color so that it can be easily distinguished in each series?

**Answer:** We chose these colors to make it easier for persons suffering from color blindness, but we agree that the result is not very good, we made new figures 1 and 2 with other colors.

**Comment:** Line 449-450. 2A and C?

**Answer:** Thank you, this is a mistake, we changed it as suggested

**Comment:** $K^{cond}$ can be written as $K^{cond}$

**Answer:** The reviewer is correct.

**Comment:** Line 461. sweater:   Seawater?

**Answer:** Yes! Thank you.

**Comment:** Line 475. Please insert the unit for [L].

**Answer:** We added nM Eq

**Comment:** Line 481. There are 3 significant digits and 4 significant digits of log K, so it is better to unify them (only the 0 in the 4th digit disappears?). I think this can be said for the entire manuscript and tables. Or is it due to a difference in method (TAC or SA)?

**Answer:** Yes we agree and changed all logK$^{cond}$ values into values with one digit.

**Comment:** Line 489. Buck et al. (2007)?

**Answer:** 2010. Thank you for pointing out this mistake, the 2010 is in the reference list.

**Comment:** Figure 6. The resolution of the figures is rough.

**Answer:** We improved the resolution.

**CC1 comments Peter Croot**

**Reviewer:** This is an interesting manuscript and a welcome addition to this field, however in reading through its conclusions regarding the underestimation of the model ligand concentrations it is apparent that some key aspects have been overlooked in the analysis to date.

**Answer:**Thank you Peter Croot.

**Reviewer:**

1. Independent determination of ligand concentration

For many of the siderophore ligands the purity of commercial sources is not 100% and for desferrioxamine B it is typically 90-95% depending on the manufacturer and lot/batch number. Earlier works also suggested that some siderophores were not stable in solution as they were easily degraded, though other studies have shown that solutions can be stable for days to weeks (Hayes et al., 1994). It is important then to obtain an independent determination of the model ligand concentration, rather than simply assume 100% and using the mass weighed out initially. This is frequently done using ASV in clean KCl or NaCl solutions and titrating with Cu, as then there should be no interferences. In the current work there is no information on how the ligand concentration was assayed prior to analysis.

**Answer:** This is an interesting point raised by the reviewer. We did not estimate the purity of the siderophores. A deviation from 100% purity of the siderophores can explain the deviation of [L] from the added amount. However, it does not change the differences between the applications and the discussion on this point. We added in the text that we did not do any research on purity of the siderophores:
At the methods section second line of 3.1.4 lines 238-240: "No tests were undertaken to check the purity of the siderophores. The solutions were used within two weeks after preparation, and kept in the refrigerator in the dark at 4∘C, which should at least for DFOB be short enough to prevent degradation (Hayes et al., 1994)".

In the discussion we added at lines 525-528 (accepted version): "It is possible that the siderophores used are not of 100 % purity, which would result in a systematic underestimation of L. However, whilst it is interesting to note absolute values, our research focusses on differences between the three applications, which is should not be impacted by any impurities."

**Reviewer**
2. Composition of seawater used in this study:

The paper states that a range of leftover samples were used in this study, though there is no information on their salinity or potential for having other metals which may complex the model ligands under the experimental conditions. For example Cu and V may also be present in seawater at significant concentrations to chelate DTPA, siderophores and fulvic/humic acids thus resulting in lower than expected ligand concentrations when titrated with iron. The question then is, which metals could be present under these conditions to outcompete iron for the model ligands tested? This also is a reminder that all

measurements done in natural waters are conditional measurements and this applies to the ligand concentration as well as the stability constant.

**Answer:**
The reviewer is right, we should have added information on this point. All water used was from the Atlantic Ocean, surface waters were not used. The same water was used per experiment for the three applications, thus the competition between metals for the model ligand should have been equivalent across all three methods. However other metals could have influenced the results. This should have been discussed. As you will see below in our answer, we used part of your text in the addition to the manuscript.

For your information below some info on the samples taken for ligands at depths>100m. We do not give this info in the manuscript, since it might be misleading, we do not know which samples were used, above all they were mixed and for every treatment the three applications received sample from the same UV irradiated mixture:

| Parameter | average | stdev | unit | N |
|---|---|---|---|---|
| pH | 7.93 | 0.08 | | 370 |
| DFe | 0.64 | 0.41 | nM | 434 |
| DMn | 0.23 | 0.18 | nM | 434 |
| Salinity | 35.09 | 0.61 | | 434 |

We added in the method section at fourth line of section 3 Methods (line 186), after:"Consequently, one batch differs from others with respect to DFe content":
", and also potentially in other constituents, such as other trace metals. Since surface samples were not used we do not expect large differences in salinity, the average salinity was 35.09 ±0.61 (N=434), obtained as average of all samples >100 m depth taken for the ligand analysis in Gerringa et al. (2015)."

We added in the discussion section at 4.2 Titrations Line 424: "Differences due to variations in sample materials are assumed to be small. However, a variance in the content of metals that could compete with Fe for ligand sites can have influenced the results and might have caused an underestimation of the model ligand concentration and indirectly also have influenced the value of $K^{cond}$. This could not have influenced the comparison between the applications, since always the same mixed sample was used per experiment for the three applications. We again emphasize that CLE-AdCSV titrations in natural waters result in the derivation of conditional parameters and this applies to the ligand concentration as well as the stability constant"

**Reviewer:**
3. Phytic acid is not a strong iron chelator under seawater conditions:
While the earlier study on Phytic acid by Witter et al. (2000) suggested that this ligand was capable of chelating Fe(III) in seawater, subsequent work suggests this isn't the case. Indeed calculations based on thermodynamic data (Crea et al., 2008; Torres et al., 2005) suggest that no significant complexes would be formed under seawater conditions. Voltammetric studies (Marolt and Pihlar, 2015) do indicate that both Fe(III) and Fe(II) complexes are formed however though they are very weak. Ultrafiltration studies (Schlosser and Croot, 2008) also indicate that the conditional binding constants in seawater for Phytic acid are significantly lower than that reported in the kinetic titrations of Witter et al. (2000). While Purawatt et al. (2007) using FFF found that Phytic acid reacts with Fe(III) to form colloidal material. These results suggest that Phytic acid is not a strong iron chelator

(Luther et al., 2021) and the results reported in the current manuscript should be reinterpreted along those lines.

**Answer:**
Thank you for this useful information, we indeed should have elaborated on this, since our estimations of $K^{cond}$ by all methods (except one duplicate obtained with TAC and this value has a large error) are lower than the values of 22.3 (with respect to $Fe^{3+}$) given by Witter et al., 2000.
We added at section 4.2, line 516:
"However, other research reported lower $K^{cond}$ for Fe-phytic acid complexes. Schlosser and Croot (2008) combined crossflow ultrafiltration with the Fe radioisotope (55Fe) and obtained a substantially lower value (18.6 with respect to $Fe^{3+}$). Moreover, phytic acid can form colloids with FeIII (Anderson, 1963). Colloid formation will interfere in several ways with the analysis by the loss of Fe, since formation of colloids results in a potentially inert fraction of Fe, the loss of phytic acid, and interference of the colloids on the mercury electrode surface. However, the formation of these colloids is dependent on the phytic acid concentration (Rijkenberg et al., 2006; Purawatt et al. 2007) and at 2 nM phytic acid we do not expect colloids to be formed."

**Reviewer:**

Lastly a recent paper (Sanvito and Monticelli, 2021) has suggested that pH buffering is not required for measurements such as this though despite earlier works indicating that it is a critical parameter. One aspect where all speciation work could be improved, and the current work should be no exception, is to include the relevant information on the pH scale (Dickson et al., 2016) being used (NBS, total, seawater, free) to describe the system, along with temperature and salinity (ionic strength) to fully describe the experimental system.

**Answer:**
We are also convinced that pH is a critical parameter. This applies for the actual measurement as well as the natural conditions (Avendano et al., 2016; Gledhill et al., 2015; Ye et al., 2020; Zhu et al., 2021). We added the pH scale we used, thank you for pointing out this omission. We used the NBS scale added now at the third line of the section 3.2 AL calibration.

author Luis Laglera gave a comment on the paper of Sanvito and Monticelli:
What Sanvito and Monticelli do is leave the pH drift exclusively at the limit layer of the electrode (microns thick) exclusively during the potential scan. In this period H2O2 and OH- are formed at the electrode surface as a result of the oxygen reduction reaction (half wave potential about -0.1 to -0.2 V). Their solution bulk pH is controlled by the natural carbonate buffer since they do not purge, this buffer is not affected. What they claim is that this substantial increase of pH during the few seconds of the voltammetric scan increases the sensitivity. This is something Luis checked personally.
Since the amount of OH- formed is so small, the pH of the limit layer can go to 9 (which can be determined by the drift of the peak potential) but the pH of the bulk of the solution remains constant leaving the sample unaffected (checked with a pH electrode inserted in the cell) (Laglera et al 2016). So, the pH is stable up to the quiescence period and then only the tiny percentage of complexes adsorbed onto the electrode which complexed iron is going to be reduced experience a rise in pH for a few seconds.

**Answer** references:

Avendaño, L., Gledhill, M., Achterberg, E. P., Rérolle, V. M. C., and Schlosser, C. Influence of ocean acidification on the organic complexation of iron and copper in Northwest European shelf seas; a combined observational and model study. Front. Mar. Sci. 3, 58, 2016 doi:10.3389/fmars.2016.00058.

Gerringa, L.J.A., Rijkenberg, M.J.A., Schoemann, V., Laan, P., de Baar, H.J.W. Organic complexation of iron in the West Atlantic Ocean. Mar Chem. 177:434-446 .doi.org/10.1016/j.marchem.2015.04.007, 2015.

Gledhill, M., Achterberg, E. P., Li, K., Mohamed, K. N., and Rijkenberg, M. J. A. Influence of ocean acidification on the complexation of iron and copper by organic ligands in estuarine waters. *Mar. Chem.* 177, 421–433. doi:http://dx.doi.org/10.1016/j.marchem.2015.03.016, 2015.

Laglera, L.M, Caprara, S., Monticelli, D., 2016. Towards a zero-blank, preconcentration-free voltammetric method for iron analysis at picomolar concentrations in unbuffered seawater. Talanta 177, 421–433. doi:http://dx.doi.org/10.1016/j.marchem.2015.03.016.

Ye, Y.; Völker, C.; Gledhill, M. Exploring the Iron-Binding Potential of the Ocean Using a 691 Combined PH and DOC Parameterisation. *Global Biogeochem. Cycles* **2020**. 692 GBC20978. https://doi.org/10.1029/2019GB006425.

Zhu, K. Hopwood, M.J., Groenenberg, J. E. Engel, A., Achterberg, E.P. and Gledhill, M., 2021.Influence of pH and dissolved organic matter on iron speciation and apparent iron solubility in the Peruvian upwelling region. Environ. Sci. Technol. 2021, 55, 13, 9372–9383

**Reviewer:**


**CC2 Randelle Bundy**

Below copied comments from R. Bundy. In between her text (**Reviewer**) our answers are given (**Answer**)

**Reviewer:**
**Comments on "Comparing CLE-AdCSV applications using SA and TAC to determine the Fe binding characteristics of model ligands in seawater"**
The authors present a very careful and interesting study that compares some of the most common CLE-AdCSV techniques used to measure the organic speciation of dissolved iron (dFe) in seawater. These experiments are very painstaking to do, so I commend the authors on such an undertaking.
**Answer**:
Dear Randelle Bundy.
Thank you for your reaction on our manuscript. Your comments, below, are interesting and help to improve the discussion on the voltammetric applications.

**Reviewer:**
Overall, I found the comparison between techniques using model ligands an important contribution to the field, but unfortunately it appears that there might have been some major issues with the application of the SA methods particularly in the kinetic experiments (e.g. Figure 6). On a broader note, I also believe that this paper is demanding more than can be expected out of these electrochemical methods. These voltammetric approaches have always been operationally defined (as all seawater iron analysis methods are) and it has been known for some time that applying these methods at different analytical windows will give you a methodspecific perspective on the continuum of iron ligands in seawater.

**Answer:**
We agree fully that the methods are operationally defined and we also prove this in our manuscript. Indeed, the detection window is one of the criteria with which we can explain differences between different applications. However, therefor we must know the detection window, and in order to calculate the detection window, we must know which Fe-AL species are formed and which are electro active.
We wish to understand factors that control Fe biogeochemistry in the ocean, and in order to do so it seems reasonable to expect that the methods used to do this 1) obey assumptions required for the data transformation and 2) produce comparable results with minimum uncertainty and 3) resolve quantitatively the simple case, of a single ligand present. If the methods we use for our investigations do not comply with these requirements we have to ask whether what we are doing is worth the effort and time required. In our study we analyzed model ligands, in an effort to see if the methods applied comply with these three criteria. Ultimately our results suggest we obtain numbers that can neither be compared between techniques and also provide dubious quantitative information about the research question. Since results were sometimes higher or sometimes lower than expected, we have to highlight at least that the uncertainty in results is probably much higher than the uncertainty obtained just from curve fits. Where does this lead interpretation of what are typically quite minor changes in L and K in global ocean data sets?
We started the experiments for two reasons. Firstly, already for some time there has been a discussion about standardizing the ligand analysis methods, and in fact this is also an important

component of the GEOTRACES program. Our manuscript could be a start. The second reason was coming from the work of Slagter et al, (2019). Slagter et al., compared TAC and SA in the Arctic. We knew TAC was not performing very well when humics are present, so we wanted to use SA. However, we also wanted to connect the future with SA to our past with TAC. So we applied both to a part of the Arctic samples. We then saw a more or less constant ratio between the obtained [L], without any relation to the presence of the humic rich transpolar drift. That was surprising. Our co-author Indah Ardiningsih extended the comparison by comparing TAC SA and NN. Again the results were puzzling.

**Reviewer:**
Bruland et al. (2000)
highlighted this very eloquently, and although that intercomparison applied to Cu speciation, its findings are applicable to Fe speciation as well. The main goal of these CLE-AdCSV methods has never been to quantify a specific ligand's concentration and binding strength, but instead to get a broad picture of the multitude of ligands that are capable of binding Fe in seawater, and to qualitatively evaluate this in seawater where a continuum of ligands exists.

**Answer**:
We do not agree with the above. There are many publications that split the ligands into more than one fraction or describe ligands as "weak" or "strong", even assigning the nature of the natural ligand based on the comparison of K from natural ligands and model ligands as we did in this work. All these interpretations are dependent on the idea that L and K can be reliably quantified with low uncertainty. Indeed, this kind of splitting was recommended in the review of Gledhill and Buck (2012), but our study suggests such splitting could result in over interpretation of data. The issue is likely less significant for Cu than Fe since the excess ligand concentration is typically larger for Cu, Cu is quite soluble (and shows much less affinity for cell walls, mercury drops etc.) allowing for much greater confidence that equilibrium can be achieved.

**Reviewer:** From studies where
these methods have been applied on the basin scale, we have gained unparalleled insights into Fe and ligands dynamics in the ocean, that are robust and oceanographically consistent and are able to be captured in global biogeochemical models (Tagliabue et al. 2017).

**Answer:** Of course we accomplished a lot (Here we mean with 'we' and below 'our' not the authors and the reviewer but the whole community working on the subject.). But that does not mean we cannot improve. Moreover, as far as we know Tagliabue et al. do not use our data because it does not fit their models. Moreover, even if a specific model converges to a solution does not mean that the data are robust. This must be ascertained with experiments as the one shown here, where a simple system must be perfectly resolved. Modellers are still struggling to fit DFe in the global models.
As an illustration we quote Ye et al., 2020:
"Global ocean biogeochemical models consider the organic complexation of iron with different degrees of complexities (Tagliabue et al., 2016). While some still assume constant ligand concentrations, others have been trying to capture the spatial variability of organic ligands with different assumptions of their relationship to biological activities. For example, Misumi et al. (2013) linked ligand concentration to oxygen consumption, and Tagliabue and Völker (2011) considered an empirical relationship between total ligand concentration and DOC. In common with other global iron models, we previously described

the role of iron-binding ligands **in the iron cycle with a constant ligand** (Ye & Völker, 2017) and with a **prognostic ligand assuming a source of ligands from DOC** (Völker & Tagliabue, 2015). Although surface DFe distributions in these models encompassed a similar range to observed DFe concentrations, there were some systematic discrepancies: DFe concentrations are generally too low in the Pacific Ocean, and the models could not reproduce the observed inter-basin pattern in the deep ocean when keeping the range of concentrations close to measured data (for details, see section 3.1.4). These discrepancies indicate that some processes controlling the spatial variability of ligands and DFe are still likely missing in biogeochemical models of iron."

**Reviewer:** Thus far, these are the
only methods we currently have to learn about the Fe-binding ligand pool as a whole. Other methods exist to measure specific ligands and functional groups, and what we have learned from those techniques is thus far unable to help us explain global distributions of dFe. It would be a shame in my opinion, to not comment on the valuable insights CLE-AdCSV techniques have brought us over the years, and to reduce the findings of this paper to a take home message that these methods are fatally flawed and we need new methods for measuring organic Fe speciation. Innovation in techniques is always a good idea, but I worry about how this paper might impact the direction of the field. Given the stature and expertise of the authors, I would have appreciated some discussion of the strengths of CLE-AdCSV methods and the instances of where and when they do work, and where and when they should be applied with caution and pause. I have outlined some of my other major comments and thoughts below, and have noted some additional specific comments at the end of the document.

**Answer:** We agree that the valuable insight the method brought us is a good addition to the introduction.
In the introduction we added at lines 64-68:
"The application of this method enlarged our knowledge on the marine chemistry of Fe and the results formed the base of the explanation why DFe depth profiles deviated from those of other trace metals (Van den Berg, 1995; Rue Bruland, 1995; Hutchins et al., 1999; Croot et al., 2004; Laglera and van den Berg, 2009; Gledhill and Buck, 2012; Bundy et al., 2014; Buck et al., 2015; Hassler et al., 2019; Lauderdale et al., 2020)."

We added in section 4.2-Overall, lines 641-643:
"In future use of CLE-AdCSV careful consideration is needed for the interpretation of the obtained [L] in relation to the application and environmental variations in ligand groups, especially the humic substances."

We did suggest how to continue with CLE-AdCSV, we end in the conclusion with the text:
At line 781: "In case voltammetric methods are used, we advise the SA5-application but to be careful to draw conclusions from obtained $K^{cond}$-values, since apart of the constraint forced by D, these estimates have a much larger error than apparent from those obtained from the fits to the Langmuir isotherm."

Still one cannot turn a blind eye to results that show the technique limitations.

**Reviewer:**
**General comments**
Kinetic experiments and the loss of signal with the SA method: The authors mention that all vials

were conditioned overnight prior to the start of experiments, but in my lab, we have found that thorough conditioning for usually more than one week is absolutely required for accurate titrations and optimal sensitivity with the SA method Abualhaija and van den Berg, 2014).

Answer:
We did not condition only overnight, we repeated overnight equilibration five time, hereafter a copy of the text from methods '3.1.2 Conditioning and equilibration', at lines 214-216:

"Before use, all materials such as vials, bottles and cells, were conditioned overnight with the prepared combinations of seawater and ligand. The conditioning procedure was performed at least three times for the analysis with TAC and at least five times for analysis with SA."

We are aware of the stricter conditioning before using SA. Our materials, equipment, all was divided into SA and TAC use. We are familiar with the SA methods and applied SA25 and SA5 and published the results (Slagter et al., 2019, Ardiningsih et al., 2021)

**Reviewer**: We are not sure why this is the case with SA, but we have observed it repeatedly as we purchase new Teflon vials and condition them.

Answer: Teflon is very reactive, aggressive cleaning before use, will mean a thorough conditioning is required. Only then there is no net flux of ligands and metals from/to the material surface.

**Reviewer:** As most titrations that we perform in the lab take several hours to complete and no SA signal loss is observed over that time frame, the dramatic loss of SA signal in the kinetic experiments after only minutes in some cases, is alarming. We also routinely perform overnight equilibrations with SA at 5, 10 and 25 μM in my lab with no loss of signal. Was the effect of conditioning ever tested with SA in this study? It is possible that the formation of a non-electroactive $Fe(SA)_2$ complex at higher SA concentrations (and also higher Fe concentrations) is related to this conditioning issue. For example, we have found that adding high Fe concentrations to our Teflon vials along with buffer and the appropriate SA concentration yields the best conditioning results, and no resultant loss in signal. Perhaps this might be because $Fe(SA)_2$ is formed under the conditions of the conditioning, and this complex has different adsorption properties to Teflon than the Fe(SA) species. The ultimate reason for the conditioning is unclear to me, however it is clear that with proper conditioning no loss of SA signal should be observed over time in the vials. I am worried that the results of the kinetic studies and even the model ligand results are over shadowed by these potential conditioning issues, rather than a lack of equilibrium of Fe with SA.

Answer:
We only succeeded in doing overnight equilibration at low [SA] of 5 μM. At 25 μM we had to follow the procedure of Buck with 15 minutes waiting time between the addition of SA and the measurement. A procedure described in the literature, also in your work (2014, 2015, 2016). If $Fe(SA)_2$ formation and adsorption on the teflon walls is indeed the reason of the need of conditioning, this indicates that research is needed, since the formation of $Fe(SA)_x$ species will shift the center of the analytical window (D).. Without proper knowledge of D and of which

species are electroactive it is unclear how to interpret results. However, adding more Fe will not favor Fe-SA species with a coordination SA>Fe, but the reverse, it will favor species with a coordination SA<Fe.

**Reviewer:**
The authors discuss the impact of mercury drops in the titration cell on SA measurements and first say that the mercury should have no effect (since they did this experiment also on a Metrohm with smaller mercury drops) and that the drop in signal is instead due to a disequilibrium because of the formation of Fe(SA)2. They then seem to contradict this idea later in the manuscript. Accumulation of mercury in the titration cell is a big problem for SA measurements, because the adsorption potential for Fe(SA) is 0V. Thus, uncharged mercury in the bottom of the cell can adsorb mercury and thus competes with the "active" drop for the binding of Fe(SA). The other issue is that the accumulation of mercury in the bottom of a cell, for a BASi instrument in particular (where a stir bar is used instead of a suspended rod), can also physically impede the stirring process, which then dramatically decreases the sensitivity (peak height) of the measurement. The bottle versus in-cell kinetic experiments perfectly illustrates this, and yet the discussion of these results is largely presented in the context of the disequilibrium argument for SA. I think the authors should make this result very clear in the manuscript, and highlight that this observation is a not a definitive support of disequilibrium. I also do not really understand the goal of the experiments where the mercury "puddle" is placed into fresh seawater and then no additional Fe(SA) signal is observed. I may have misunderstood how this experiment was performed, but based on how it is written in the text I do not think a lack of signal in this experiment signifies irreversible formation of Fe(SA), but instead reflects the fact that Fe(SA) adsorbs at 0V and is not reversibly removed until a negative potential is applied to the mercury drop(s).

**Answer:**
The idea that a mercury electrode at 0V is equivalent to a mercury drop at the bottom of the cell is not correct. A cell held at 0V (vs the reference) holds a potential and a surface charge. The potential of zero electrode charge for a polarographic instrument with an Ag/AgCl reference electrode is in between -0.4 and -0.5 V in seawater. The continuous decrease of a peak cannot be assigned to the effect of the Hg puddle or the stirrer, otherwise, it would be impossible to measure duplicates and triplicates of a sample.

About the mercury puddle, we agree that our experiments did not give a clear proof of what happens with Fe-SA species at mercury drops at the bottom of the cell. Adsorption on the mercury at the bottom of the cell was indicated by Buck et al. (2007) to be the reason of a decrease or instability of the signal. However, this was contradicted by Abualhaija and Van den Berg (2014) who argued that the changing conditions in the oxygen concentration was the reason of the instability, since samples were not purged only stirred with the SA method using BASi equipment. That is why we checked, our results are inconclusive as said at the end of point 3 page 26, line 713, section in-cell kinetics: "It remains hard to explain the strong reduction of the peak height without a relationship with the mercury volume at SA5"

But it is obvious that discrepancies among authors justify our investigation,

We concluded the in-cell kinetics section with:
In-cell kinetic experiments with SA did not reach equilibrium and showed a continuous decay of the peak height. This can be explained by a combination of processes like adsorption of Fe(SA)-complexes on

dispensed mercury at the cell bottom, and formation of the irreversible (FeSA+SA forms $FeSA_2$) and according to Abualhaija and Van den Berg (2014) to the non electroactive $Fe(SA)_2$.

Here it is important to understand that FeSA adsorption onto Hg (electrode or puddle)  may also be driven by hydrophobicity.

**Reviewer:**

Comparison of the TAC and SA methods with model ligands: I found this section very interesting, and it is an important aspect of this work to report to the community. However, the model ligand section is hard to follow because it is not always clear what the measured values are being compared to in terms of the expected or "true" value. The expected values of each model ligand based on what has been seen previously in the literature would be immensely helpful to include in Table 2. Also, please label each logK$_{cond}$ with either a Fe3+ or Fe' subscript, because I was often getting confused which one you were reporting in some figures, tables and in the text, particularly in Table 2. For example, you report the model A ligand logK$_{cond}$ with respect to Fe3+ and the model B ligands with respect to Fe'. Yet, in Table 2 you report the results for model B ligands with respect to Fe3+. Please keep these consistent so that comparison across methods and to previous studies are possible. I also think it would be helpful to report the "true" or expected values for each model ligand because that might also make Table 3 more meaningful. Table 3 is useful in terms of how the different methods perform relative to one another, but how about how they perform relative to what is the expected value? This would be much more insightful to consider. I actually had to make my own table while I was reading the manuscript in order to see for myself how the measured values compare to previous literature results (and thus the "expected" value for each model ligand).

==**Answer:**==

About the $K^{cond}$,

Indeed we should use consequently one way of expressing the conditional stability constant and we did, all values given are with respect to $Fe^{3+}$, with the exception of Table 1. In this table we gave both values, with respect to $Fe^{3+}$and with respect to Fe′, since the applications use a different pH and thus different inorganic alpha values were used. It is clearly indicated in the headings of the columns.

However, in Table 2 and 3 we used logK without any specification, here we now added in the table captions: "LogK is used for log$K^{cond}$ with respect to $Fe^{3+}$."

We are sorry that the table was cut off. We repaired this now. The editing team pointed it out.

About the true value:

We do know for the section A model ligands how much we added, so indeed there is a true value for [L]. This is not the case for the B model ligands, since [L] is determined in a method-specific way, and above all the molecules consist of mixtures of unknown composition. For the conditional stability constant we assumed two true values, for DTPA and DFOB (indicated in the log log plots as a dashed black line). To show how little we know of Fe-L stability constants in seawater is illustrated in this review process by the comments submitted by Peter Croot: Witter et al., 2000 published for phytic acid logK$^{cond}$=22.3, whereas Schlosser and Croot ,(2008) published a value of 18.6. These values both with respect to $Fe^{3+}$ differ a factor 5000.

Since we compared applications, we prefer to show how the different methods perform relative to one another. Still figure 2 shows for two model ligands how the applications compare to the true values.

**Reviewer**

I think it is also important to thoroughly discuss the detection window being used in each method, and how that compares to the analytical window of each model ligand experiment. A quick calculation of the analytical window for each model ligand case shows that several of the titrations are likely outside of the analytical window for some of the methods. For example, based on the expected logK for each model ligand from previous literature values, the TAC method was best suited for measuring most of the model siderophores and it often performed better than SA for these model ligands. In the opposite case, the SA 5 μM method performed better with the weaker humic and fulvic ligands. Making the connection between analytical window and the model ligand being examined clear in light of the results obtained is critical.

==Answer:==

Indeed the detection window or analytical window (D) is important to interpret the quality of the results. In Table 1 we show D of the applied methods as we calculated them with our calibration. Note that both SA applications have a comparable D, even though the [SA] differed a factor 5. This is explained by Abualhija and van den Berg (2014), due to the formation of the two different Fe-SA species. The D of TAC is higher, and thus indeed should be better equipped to detect the $K^{cond}$ of DFOB. Both SA applications should be better equipped to detect the $K^{cond}$ of the humic substances, DTPA and phytic acid (if for the latter we assume Schlosser and Croot (2008) have published the correct value).

We added as introduction to section 4.2 Titrations-Overall Line 593-onwards : "The log-log plots for DTPA and desferrioxamine B, between known and observed conditional stability constants show that the data points obtained by TAC are closest to the theoretical curve of DTPA, and by SA5 are closest to desferrioxamine B (Figures 2A,C). However, the TAC application has the highest D and should therefore be better equipped to detect desferrioxamine B, and least equipped to detect DTPA."

**Reviewer**

Recommendations for future work and insights from past work: Given the implications of this work to the field and the extensive knowledge and background of the authors, I was hoping there would be a final section of the manuscript with recommendations going forward. The authors mention that we need to find new ways to measure the speciation of Fe in seawater, but make no suggestions. The authors should also comment on how past results might be interpreted. When I made my own table where I compared the measured ligand concentrations and logKs from each model ligand study to past results seen in the literature, to my eyes there was no systematic "best" method. It was often dependent on the model ligand and the analytical window where that model ligand falls, relative to the analytical window of the method used. Some discussion that brings all of the insights from this paper together and gives recommendations going forward beyond, "we need a better way" would be very powerful. As both an electrochemist and a mass spectrometerist, I can say with certainty that no method is perfect, and each has its benefits and pitfalls.

**Answer:**

Any compilation of published methods for model ligands shows strong deviations of values as a function of the added ligand, making this work more necessary. Another important conclusion of this work is that we find that errors in the calculation of log K are higher than the usual difference of D among methods. This increases extraordinarily our uncertainty and limits extraordinarily our ability to compare values obtained with different methods.

The suggestion,of recommendations for future work is indeed a good suggestion, although a full review of the pros and cons of available alternatives is beyond the scope of this manuscript. We extended the conclusion section Lines 780-810 as follows:

"All methods have drawbacks, and the characterization of Fe-binding organic ligands at nM concentrations in seawater with a high ionic strength and at 40-80 µmol/kg DOC (Hansell et al. 2009) is and remains challenging. In case voltammetric methods are used, we advise the SA5-application, but should not be overinterpreted and issues relating to the technique fully acknowledge. In particular, it is hard to justify determination of equilibrium constants using an adsorption isotherm that assume equilibrium, if a signal is not stable and the experimental system is therefore not at equilibrium (for whatever reason), especially bearing in mind the method specifically uses the term "equilibrium" in its title. Furthermore, it appears that apart from the constraint forced by D, these estimates have a much larger error than apparent from those obtained from the fits to the Langmuir isotherm, which may explain why direct links between speciation predicted from conditional parameters do not relate strongly to bioavailability (Shaked et al., 2020, 2021) although this could also be caused by the limited role assigned to iron reduction prior to bio-assimilation. Given that the research questions that are typically addressed when determining conditional parameters from the Langmuir isotherm actually relate to how variations in the parameters impact on Fe speciation (e.g. abundance of Fe not bound to organic matter and thus assumed to be bioavailable), it would seem appropriate to broaden the methodology applied to these questions to other methods capable of e.g. estimating concentrations and distributions of the dominant groups of binding sites and/or the lability and solubility of Fe. For example, Whitby et al. (2020) and Laglera et al., (2019) have used alternative methods based on voltammetry to suggest that humic substances are more important than thought before. Other recent studies have focused on the influence of the pH on metal organic complexation and vary the pH of titrations (Ye et al., 2020, Gledhill et al., 2015; Zhu et al., 2021). Here we note that the acid-base chemistry of organic matter – which underpins metal binding - is severely understudied, but can nevertheless quantify the distribution and concentration of the total cation binding sites present, at least in the portion of dissolved organic matter that can be isolated from seawater (Lodeiro et al., 2020). With further knowledge of acid-base chemistry of dissolved organic matter and how it changes with inputs of fresh material from biological activity, the total binding site concentration could be independently constrained and only binding affinities derived. Alternatively models can be applied that allow for the estimation of metal speciation for ambient conditions (e.g. Hiemstra and van Riemsdijk, 2006; Stockdale et al., 2015), with predictions of how binding to organic matter could influence e.g. iron solubility (Zhu et al., 2021). The wider application of methods employing cation chelating resins, such as diffusive thin film gradients (Zhang and Davison, 2015; Town et al., 2009; Bayens et al., 2018) might also offer alternative insights into the lability of metals in seawater. Another way to apply voltammetric methods is to characterize metal-binding ligands by pseudovoltammetry (Luther et al, (2021), although unfortunately this method is not currently suitable for DFe. Whilst continued application of the CLE-AdCSV approach will no doubt further develop our knowledge on how operationally defined ligand concentrations and stability constants vary in the ocean under a restrictive set of conditions apparently specific to one added ligand, new approaches to both the

determination and the interpretation of metal binding to organic matter will surely stimulate discussions in the field of the organic metal complexation and furthermore be likely lead to new insights.

**Reviewer:**
**Specific comments**:
There are several small typos, only some of which I have detailed here.
Section 2: You list your assumptions and refer to them by number, but they are not numbered. Numbering them might be helpful, since this whole section reads like a list and you refer to specific assumptions later in the manuscript.
**Answer:** Thank you this is repaired.

Line 104: Add a space between "knowledge" and "In"
**Answer:** Thank you this is repaired.

Line 108: Do you mean 20 minutes or 15 minutes? You use 15 minutes throughout the manuscript, and say that you also used a timer.
**Answer:** Thank you this is repaired into 15.

Line 129: Add a space between "[L]" and "and"
**Answer:** Thank you, this is now adapted.

Line 181: Were the samples filtered prior to being stored frozen? If so, how were they filtered? Where they frozen at -20C?
**Answer:** we changed the first sentence into (line 183): "The natural seawater used in the experiments consisted of mixed leftover filtered samples of the northern Western Atlantic GEOTRACES cruise GA02 (stored frozen at -20 ∘C) (Rijkenberg et al., 2014)."

Line 182: Was the UVSW aged prior to use or used immediately?
**Answer:** Used without aging, as shown by" UV irradiated sea water was stored for 3 days at most."

Line 215: Was all kinetic conditioning done in UVSW?
**Answer:** We added: in UV-irradiated seawater

Line 225: Remove the second "in" after "placed"
**Answer:** Thank you, this is now adapted.

Line 227: Do you have a reference for this?
**Answer:** Yes we changed the text already due to remarks of other reviewers and added Rapp et al., 2017; Wuttig et al., 2019).

Line 260: Bundy et al. (2018) determine the conditional stability constant of ferrioxamine E in seawater (as well as ferrioxamine B). The $\log K_{=14.4}$
$_{\%\&}$ for ferrioxamine E was 14.05 and the $\alpha_{!"!}$
used was 10. These measurements were performed using the SA method at 5 μM.
**Answer:**
Apologies for missing this. We added this information now in the text at line 271-273:

But Bundy et al. (2018) estimated $\log K_{FeL,Fe'}$ values of ferrioxamine B and E , using SA5 in seawater to be close being 14.4 and 14, respectively.

Line 308: Were blanks also in absence of Fe?
**Answer:** No, the blanks were UV irradiated seawater for the Atlantic Ocean, so these contained approximately 0.64±0.4 nM DFe (N=434) (obtained from all samples <100 m depth described in Gerringa et al., 2015).

Line 313: This sentence is confusing. I think you mean that you added buffer and dFe and not also SA, and then you added SA after equilibration. Why equilibrate the buffer and dFe for one hour? The commonly used method equilibrates the buffer and dFe for two hours before adding the SA (Buck et al. 2007).
**Answer:**
We changed "the 25μM SA buffer" into: "the buffer".  This was indeed confusing.

Line 321: I think you mean "prior to the addition of TAC or SA."
**Answer:**
We changed the sentence into: "In the latter case, there is the possibility that Fe-oxide precipitates were formed prior to the addition of TAC or SA and were probably dissolved after the addition of TAC or SA".

Line 357: Why was this dFe concentration chosen? Was your seawater a deep sample? Why not used the measured dFe concentration?
**Answer:**
The measured DFe varied per sample. Our mixed Atlantic seawater was >100 m depth. To answer reviewer Peter Croot we calculated the possible average DFe from all samples taken for ligand characterization to be 0.64 nM. However, we do not know which samples were used, they were leftovers. Anyhow it is not important for the calculations which DFe was used, as long as it is the approximately the same magnitude as seawater concentrations.

Line 369-370: In most SA CSV studies the blank is zero, meaning no Fe is added with the buffer or SA addition. Can you include the ordering information for the boric acid and SA that were used? Was distilled or Optima ammonium hydroxide used for the buffer preparation? The fact that there was a blank with these measurements is very disconcerting. On a related note, please note the error associated with the dFe measurements for the model ligand results. In some cases, the standard deviation on the concentration of L and the logK is relatively small, and given that the dFe in each experiment varied quite widely, it would be nice to see the error on these measurements as well.
**Answer:**
We are not sure whether the reviewer refers to the voltammetric signal or ICPMS analysis. With ICPMS analysis we do not often encounter zero for DFe in solutions or samples. This is a general feature when using reagents that cannot be produced at ultraclean levels: borate, good buffers, artificial ligands and some model ligands. Residual Fe concentrations must be always checked in all reagents, otherwise, small deviations in the concentration of Fe in solution may produce strong deviations in L and K.
We had no information about the Fe content of SA, but we do know that the Fe content of TAC varies and can be too high, and that we have to discard certain batches. That is why we did the analyses. Yes we used high purity ammonium hydroxide (Merck supra pur), added now in Table S1).

See Table S1 for SA, Acros organics.

We added at section 3.7 ICPMS analyses: at line 375
"For the analyses of DFe in UV irradiated seawater with and without added model ligands the limit of detection for DFe was 22 pM±8 pM. The DFe of model ligands DTPA, Phytic acid, desferrioxamine B and ferrichrome were in the dilutions used (2 nM and 4nM) below the detection limit. This means the addition of the model ligand did not increase the DFe in the UV irradiated seawater. Analyses of 2 nM vibriobactin resulted once in <detection limit and once in 0.1 nM DFe. However, ferrioxamine E, FA and HA contained measurable amounts of DFe. 2 nM FerrioxamineE= 1.76±0.04 nM (N=2), 0.2 mg FA= 1.1 ±0.02 nM  (N=3), 0.2 mg HA=3.39±0.05 nM (N=2)."

Line 410:
Change to "ligands like siderophores point to biases"
**Answer:**
We changed the sentence by removing a confusing and wrongly placed ",".

Line 588: I think it is difficult to make too many assumptions on the HS and FA results, because there is large variability in the literature with respect to the available dFe binding sites. Again, what is the approximate "true" value we can compare these model ligan titrations to? Did you determine your own dFe binding capacity measurements for the batches of FA and HS you used?
**Answer:**
We agree, there is no true value here. Still Laglera et al 2009 showed that CLE-AdCSV and straight Fe titration of the HS until the end of a linear response gave close results. Luis Laglera did determine the binding capacity for FA (Sukekava et al., 2018), we used the same batch. For HA, the binding capacity was measured in 2009 and re analyzed recently in a different batch (unpublished results, Laglera) with a consistent result slightly over 30 nmol Fe (mg SRHA)-1.

Line 645: You can still have mercury adsorption on the drop even with smaller drop sizes. The drop on the Metrohm itself is smaller, therefore having the same sized drops in the bottom of the cell, relative to the active drop, will still give you the same issue.
**Answer:**
Correct we changed the sentence in (line 678): "Possible contributions due to decreasing oxygen were excluded , and due to adsorption on mercury on the cell bottom were minimized"
Table 1: Some of this table is cutoff, so it is difficult to understand what is being displayed.
**Answer:**
We are very sorry that this happened. We made a new table.

Randie Bundy

[revised manuscript text omitted]